# Wire Tool Electrode Behavior and Wear under Discharge Pulses

**Sergey N. Grigoriev** [1] , **Marina A. Volosova** [1] , **Anna A. Okunkova** [1,*] , **Sergey V. Fedorov** [1],
**Khaled Hamdy** [1,2] , **Pavel A. Podrabinnik** [1] , **Petr M. Pivkin** [1] , **Mikhail P. Kozochkin** [1] and
**Artur N. Porvatov** [1]

1   Department of High-Efficiency Processing Technologies, Moscow State University of Technology
    «STANKIN», Vadkovsky per. 1, 127055 Moscow, Russia; s.grigoriev@stankin.ru (S.N.G.);
    m.volosova@stankin.ru (M.A.V.); sv.fedorov@icloud.com (S.V.F.); eng_khaled2222@mu.edu.eg (K.H.);
    p.podrabinnik@stankin.ru (P.A.P.); p.pivkin@stankin.ru (P.M.P.); astra-mp@yandex.ru (M.P.K.);
    porvatov_artur@mail.ru (A.N.P.)
2   Production Engineering and Mechanical Design Department, Faculty of Engineering, Minia University,
    Minia 61519, Egypt
*   Correspondence: a.okunkova@stankin.ru; Tel.: +7-909-913-1207

**Abstract:** This work is devoted to researching the tool electrode behavior and wear under discharge
pulses at electrical discharge machining. The experiments were conducted on the workpieces of
12Kh18N10T (AISI 321) chrome-nickel anti-corrosion steel and D16 (AA 2024) duralumin by a
0.25-mm-diameter CuZn35 brass tool in a deionized water medium. The developed diagnostic and
monitoring mean based on acoustic emission registered the oscillations accompanying machining at
4–8 kHz. The obtained workpiece and non-profiled tool surfaces were investigated by optical and
scanning electron microscopy. Calculated volumetric and mass removal rates showed the difference
in the character of wear at roughing and finishing. It was shown that interaction between material
components in anti-corrosion steel machining had an explosive character between Zn of brass and Ni
of steel at a micron level and formed multiple craters of 30–100 μm. The secondary structure and
topology of worn tool surfaces were caused by material sublimation, chemical interaction between
material components at high heat (10,000 °C), explosive deposition of the secondary structure.
Acoustic diagnostics adequately registered the character of interaction. The observed phenomena at
the submicron level and microstructure of the obtained surfaces provide grounding on the nature of
material interactions and electrical erosion wear fundamentals.

**Keywords:** erosion; tool wear; sublimation; $ZnNi_x$; explosive deposition

## 1. Introduction

The subject of electrical discharge machining (EDM) and wear of a wire tool electrode is not new,
but the physical processes that occur during processing are still not sufficiently studied [1–4]. It is
related to the absence of the possibility of visual control over the processes occurring in the discharge
gap during EDM especially for large workpieces [5,6]. Plenty of studies are devoted to the processes
related to the physical phenomena of erosion wear with various conclusions [7–11]. However, at the
industrial level, there is no solution to avoid the negative consequences of the accident wire electrode
breakage or dumping of the machined part into the working tank at the end of machining. It is
especially actual in the case of splitting two co-dependent parts—die and punch for the injection mold,
micro-gears [12,13].

An experienced operator often determines the control over processing and process conditions by the specific noise that occurred in the working area. It grows with an increase in the intensity of processing (roughing or finishing) and varies during the electrical discharge machining of materials with uneven structures—porous material, set of tubes, nanocomposites, or composites. The changes in the specific sound are especially noticeable during wire tool penetration into the workpiece and at the end of machining.

The electrical discharge machining of materials occurs in specific conditions between two electrodes. A bias increase is followed by ionizing the space between two electrodes at the moment of their approach. Dielectric breakdown by spark provokes a discharge channel that creates the conditions of low-temperature plasma with 10,000 °C that can be observed in particular conditions (forming intermetallics of $Al_2Cu$ [14,15], $ZnNi_x$ [16], $CeNi_2$ [17], burning titanium at 1200 °C [18,19]) in the form of an instant growing gas bubble surrounded the discharge channel. Then, pulses interrupt to restore the breakdown conditions for the next pulse—the bias in the gap, erosion products' washing away from the working area. The occurred conditions are close to the conditions of lightning formation [20]. The temperature in the interelectrode gap achieves high value in a microsecond [21–23].

All monitoring means can be divided into optical and non-optical—electrical and vibroacoustic. The absence of visual contact with the working zone due to its tiny sizes hampers the application of any optical monitoring means. At the same time, the existed monitoring of the electrical parameters does not provide adequate data on the effectiveness of the discharge pulses since for the modern control systems all the produced pulses in the working zone are counted as working ones when a part of them can be spending on the destruction of erosion products [24,25]. It can be grounded by difficulties that met electrical discharge machining in processing materials with threshold conductivity, uneven structure, or microstructure, and inclined surfaces with a thickness of more than 100 mm.

The vibroacoustic monitoring method does not have this kind of disadvantage, as it counts only effective discharge addressed to the material to be processed on the destruction of the surface that was recorded with the help of the accelerometers placed at the working table of the machine [26,27]. It can be an effective means for not only tool behavior investigation and its influence on the quality of the machined surfaces but also an effective means for adaptive control of electrical discharge machining in real manufacturing conditions.

Simultaneously, research of the character of tool electrical erosion wear, sublimation phenomena [28–30], and the nature of the material destruction of the machined surfaces can give additional, comprehensive, and exhausting data.

This paper is aimed at the research of electrical discharge machining by the developed diagnostic means to obtain new data on the influence of wire tool behavior on the quality of the machined surfaces; wire tool wear at roughing and finishing, the nature of material destruction under discharge pulses, and sublimation phenomena.

The research is conducted for two materials:

- 12kH18N10T (AISI321) chrome-nickel anti-corrosion structural steel that is often used in injection mold manufacturing;
- D16 (AA 2024) duralumin used for aviation purposes.

The work's scientific novelty is in new data on electrical erosion wear of materials, sublimation phenomena, nature and mechanism of material destruction for two types of structural materials, dependencies between detected acoustic emission and machined surface quality, and classification of the eroded surfaces of the tool.

The tasks of the study are:

1. Applying the vibroacoustic means for research on dependencies between the tool behavior and surface quality;
2. Research of the tool and machined surface morphology and chemical composition;

3. Classification of the observed defects and traces of tool destruction at roughing and finishing of two material types;
4. Analyses of the chemical interactions between components that occurred in the discharge gap and conclusions on type material destruction and changes occurred at surface and subsurface layers.

## 2. Materials and Methods

### 2.1. Equipment

A four-axis wire electrical discharge machine, Seibu M500S, was used in the experiments for research of wire tool behavior and wear under pulses. The main characteristics of the machines are presented in Table 1.

**Table 1.** Main characteristics of wire electrical discharge machines used in experiments.

| Characteristic | Value and Description |
|---|---|
| Max axis motions along the axes X×Y×Z, mm | 500 × 350 × 310 |
| Max angle of conical machining, degree | ±10° |
| Max weight of workpiece, kg | 800 |
| Accuracy of positioning along the axes, μm | ± 1 ÷ 2 |
| Achievable roughness Ra, μm | 0.4 |
| Dielectric medium | Deionized water |
| Machine body | The frame is made of gray cast iron having good thermal and vibration compensating characteristics |

The machines are located in a thermo-constant room to reduce ambient temperature's effect on the results of processing. Workpieces were immersed in a dielectric for 10 min before processing to avoid dimensional fluctuations related to the difference in temperatures between the environment and working fluid. The dielectric height was established at $1 \div 2$ mm above the workpiece. The upper guide of the machine was placed at a minimum distance above $2 \div 5$ mm from the dielectric [31–33].

The tool electrode is a brass wire with a diameter $d_w$ of 0.25 mm made of CuZn35 (Cu—65%; Zn—35%) with a processing temperature of 260 °C and annealing temperature of 425–750 °C.

The choice of the electrode type was made since a brass tool of 0.25 mm in diameter is the most widespread for wire electrical discharge machining and suitable for the broad field of applications when the forced choice of any other electrode is due to a severe technological need and is associated with the need to purchase and reinstall expensive nozzles.

It should be noted that the positive polarization of the workpiece and negative polarization of the tool electrode is traditional for the electrical discharge machining. However, modern machine tools can switch the electrodes' orientation for some particular modes or even during machining uneven and hard-to-machine materials in automatic mode.

A CNC program was prepared manually; path offsets were not taken into account. The EDM-factors were chosen using recommendations mentioned in previously conducted works and developed by other scientific groups [25,34–36] (Table 2). The maximum working voltage $V_o$ varied in the range of 40–70 V with a pitch of 10 V for characterization of the discharge pulses by for oscilloscope research and to provoke the conditions of wire breakage for microscopic research. At least five samples and cuts were produced for each group of parameters.

**Table 2.** Electrical discharge machining (EDM) factors during experimental work.

| Factor | Value |
|---|---|
| Seibu M500S [1] | |
| Operational voltage in the interelectrode gap before the approach of the tool electrode to the workpiece, $V_o$ | 55, 60, 65 |
| Auxiliary voltage that occurs at the moment of discharge between the tool electrode and workpiece, $V_g$ | 32 |
| Strength of the working current in the interelectrode gap, $I$ | 8 |
| Auxiliary current to increase the cutting efficiency when the circuit is turned back on, $N_s$ | 43 |
| Time of disconnection of the current source, the percentage ratio of the time of the discharge pulse to the time of its absence, $T_{off}$ | 6 |
| Time intermittent pause to ensure the stability of the processing process, $A_d$ | 305 |
| Speed of the tool rewinding, $W_s$ | 35 |
| Feed speed, $S_g$ | 5 |
| Wire tension, $W_t$ | 30, 35, 40 |
| Dielectric pressure in nozzles for flushing, $F_l$ | 245 |

[1] Provided in equivalent unit of the machine.

## 2.2. Materials

The chemical composition of 12Kh18N10T (AISI 321) austenite steel is presented in Table 3; the composition of D16 (AA2024, AlCuMg2) duralumin is in Table 4. The thickness of the samples was 20 ± 0.1 mm for both materials.

**Table 3.** Chemical composition of 12Kh18N10T steel (AISI 321) in wt%.

| Element | Fe | Cr | Ni | Ti | Si | S | Mn | Cu | P | C |
|---|---|---|---|---|---|---|---|---|---|---|
| wt% | Balance | 17–19 | 9–11 | About 0.8 | Up to 0.8 | Up to 0.02 | Up to 2.0 | Up to 0.03 | Up to 0.035 | About 0.12 |

**Table 4.** Chemical composition of D16 alloy (AA2024, AlCuMg2) in wt%.

| Element | Al | Cu | Mg | Mn | Fe | Si | Zn | Ni | Ti |
|---|---|---|---|---|---|---|---|---|---|
| wt% | 90.8–94.7 | 3.8–4.9 | 1.2–1.8 | 0.3–0.9 | Up to 0.5 | Up to 0.5 | Up to 0.3 | Up to 0.1 | Up to 0.1 |

The high chromium content of the proposed in the research steel ensures the metal's ability to passivate and causes strong corrosion resistance of steel. The addition of nickel converts steel to austenite class. This property is of exceptional importance, allowing to combine the machinability with an expanded set of performance properties. The addition of strong carbide-forming element titanium eliminates the tendency to intergranular corrosion. In turn, carbon forms a refractory titanium carbide and excludes a decrease in the concentration of chromium by chromium carbides formation. It should be noted that the field of chromium-nickel steel applications dominates in the modern rolled metal market [37–40].

Duralumin D16 is a structural alloy mainly used in the aviation and space industries. D16 is rarely used in its pure form since it has less strength and hardness in the non-quenched state. The alloy is classified as a durable thermo-hardened material [41,42].

A Fischer Sigmascope SMP10 instrument (Helmut Fischer GmbH, Sindelfingen, Germany) measured the specific electrical resistance ρ of the materials used in experiments (Table 5, Figure 1a). The device measures the material electric conductance in Siemens and the percentage of the control

sample's electrical conductance produced from annealed bronze in the range of $1 \div 112\%$. All measured values were converted to $\frac{\Omega \cdot mm^2}{m}$. The melting/sublimation points of the materials $T$ provided in Figure 1b [43–46].

**Table 5.** Specific electrical resistance ρ of some materials at +20 °C.

| Material | Specific Electrical Resistance ρ [$\frac{\Omega \cdot mm^2}{m}$] |
|---|---|
| 12Kh18N10T (AISI 321) steel | 0.746 |
| D16 (AA2024) alloy | 0.028 |
| Brass alloy, CuZn35 | 0.065 |

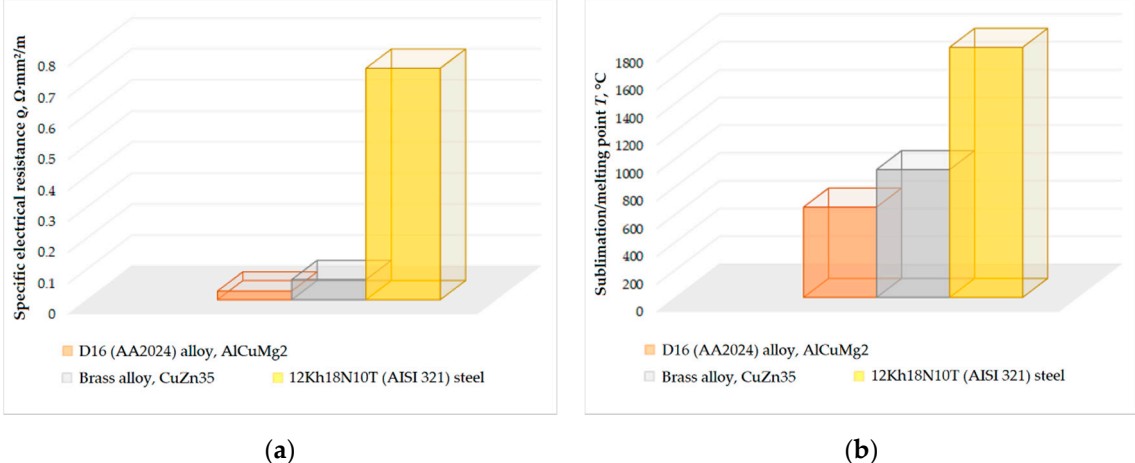

(**a**)　　　　　　　　　　　　　　　　　　　　(**b**)

**Figure 1.** Electrophysical properties of the materials used in experiments: (**a**) specific electrical resistance ρ at +20 °C; (**b**) sublimation/melting point $T$.

The stress-strain curves of chosen materials have the presence of elastic and plastic deformation zones [47,48]. The fracture formation schemes have a ductile phase that increases with the material's plasticity—from 12Kh18N10T (AISI 321) steel and brass to D16 (AA2024) alloy. The reduction area for these materials is ~$38 \div 42$ % for steels [47], ~$52 \div 53$ % for brass alloys [49], and ~$75 \div 77$ % for aluminum alloys [50]. Tensile strength $\sigma_B$ (UTS) for these materials are 510–830 MPa for 12Kh18N10T (AISI 321) steel, 450 MPa for CuZn35 brass alloy, and 345–420 MPa for D16 (AA2024) alloy.

Reduction area $S_{RA}$ is calculated by the following equation [47]:

$$S_{RA} = \frac{S_0 - S_{min}}{S_0} \cdot 100 [\%], \tag{1}$$

where $S_0$ is an original transverse area, mm$^2$, and $S_{min}$ is a minimal area of the final neck, mm$^2$.

### 2.3. Monitoring

A digital oscilloscope TDS2014B (Tektronix, Berkshire, UK) produced the characterization of the discharge pulses.

The vibroacoustic monitoring was provided by piezoelectric accelerometers mounted on the elastic system of the machine [23,25–27,51,52] (Figure 2). The data received by the accelerometers signals were forwarded to preamplifiers, amplifiers VShV003 (JSC Izmeritel, Taganrog, Russia), and an analog-to-digital converter E440 (L-card, Saint-Petersburg, Russia), and recorded with a personal computer (PC). Data were recorded at 1 min, 30 s, and 5 s before the end of processing. Spectral analysis was performed at frequencies 4–8 kHz. The signal was preliminarily cleaned from low-frequency noise using a high-frequency filter. The signal's effective amplitude was determined. The square of this amplitude is proportional to the signal that arises in the machine's elastic system under disturbing influences of the discharge pulses. The cutoff frequency of filters is 2 kHz.

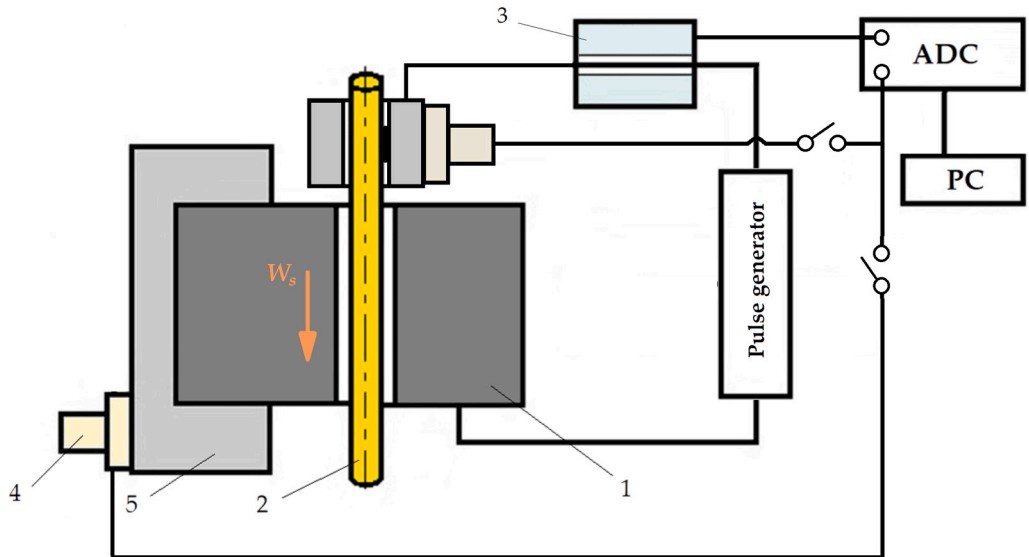

**Figure 2.** Scheme of monitoring sensor setup at electrical discharge machine: (**1**) is a workpiece; (**2**) is a wire tool; (**3**) is a current sensor; (**4**) is accelerometers; (**5**) is a workpiece fastening system; ADC is the analog-digital converter; PC is a personal computer.

### 2.4. Physical Relationship of EDM Factors and Vibration Amplitude

Typically, electrical discharge machining has a very narrow range of working factors for machining every material type. Nevertheless, up to 16 factors can be varied during machining [6]. The papers related to EDM research concentrate on some of them as we have done regarding the research subject. It was decided to vary two of the most important factors—operational voltage, which influences the density of discharge pulses distribution, and wire tension, which influences system stiffness and consequently wire oscillation amplitude. The detailed force diagram is presented in [27].

The amplitude of the wire can be presented by summarized force of working impulses in the system's action $\Sigma F_{imp}$ and stiffness $k_n$:

$$A_n = \frac{\sum F_{imp}}{k_n}. \tag{2}$$

At the same time, the stiffness of the system is determined by its mass:

$$k_n = 4\pi^2 \frac{m_n}{T^2} \tag{3}$$

where $T$ is a period of self-oscillations; thus, the signal amplitude has an inverse relationship with the weight.

Regarding operational voltage, it has a physical dependence on the signal's amplitude, since it influences the density of the discharge pulses and, consequently, the summarized force factor. The electrical impulse itself is a short-term burst of electrical breakdown voltage and working current that can be presented as follows:

$$\sum F_{imp} = I \cdot V_0. \tag{4}$$

The energy of pulses then will be:

$$\sum E_{imp} = I \cdot A_n. \tag{5}$$

The wire tension has dependence that is even more evident—$W_t$ influences the system stiffness:

$$k_n = \frac{F_e}{\Delta l}, \tag{6}$$

where *Fe* is a restoring force that is opposite and equal to the applied wire tension $W_t$ and $\Delta l$ is a change in the wire length. Thus:

$$k_n = \frac{|W_t|}{\Delta l}. \tag{7}$$

Moreover, the height of the workpiece also influences the stiffness of the system and wire amplitude:

$$k_n = \frac{E \cdot S_0}{l_n}, \tag{8}$$

where $l_n$ is a wire length, $E$ is Young's modulus, and $S_0$ is a wire area.

*2.5. Characterization of the Samples, Wear Rate, and Discharge Gap*

An EL104 (Mettler Toledo, Columbus, OH, USA) laboratory balance with a measurement range of $0.0001 \div 120$ g weighed the obtained samples with an error of 0.0001 g.

The samples' surface roughness was controlled by a high-precision profilometer, Hommel Tester T8000 (Jenoptik GmbH, Villingen-Schwenningen, Germany) with a resolution of $1 \div 1000$ nm and a measurement error of 2%.

An Olympus BX51M instrument (Ryf AG, Grenchen, Switzerland) provided the optical microscopy; the discharge gap was measured optically.

A VEGA 3 LMH instrument (Tescan Brno s.r.o., Brno, The Czech Republic) with magnification up to 1,000,000× provided the scanning electron microscopy and spectrometry of the sample.

The cross-sections were prepared according to the standard probe techniques by an ATM sample equipment—Opal 410, Jade 700, and Saphir 300 (ATM, Haan, the Netherlands). Epoxy resin with quartz sand provided pouring of the samples as a filler was used.

The worn area of the tool can be calculated by the equation of the circle segment area ($S_w$):

$$S_w = \frac{1}{2} r_w^2 \left( \frac{\pi \cdot \alpha}{180°} - \sin \alpha \right) \left[ mm^2 \right], \tag{9}$$

where $r_w$ is a wire radius, mm and $\alpha$ is an angle of the segment, degree. The volumetric ($R_v$) and mass wear rates ($R_m$) are calculated by the following equations [53,54]:

$$R_v = \frac{\Delta V}{t} \left[ mm^3 \cdot s^{-1} \right], \tag{10}$$

$$R_m = \frac{\Delta m}{t} \left[ g \cdot s^{-1} \right], \tag{11}$$

where $\Delta V$ is volumetric wear, mm$^3$, $\Delta m$ is a worn mass, g, and *t* is the wire length wear time; for a thickness of 20 mm, $t = 0.343 \pm 0.005$ s.

The discharge gap is calculated by the following equation:

$$\Delta_{DB} = \frac{l_s - d_w}{2} \left[ mm \right], \tag{12}$$

where $l_s$ is the measured width of the slot, mm, and $d_w$ is the wire diameter, mm.

## 3. Results

*3.1. Electrical Discharge Pulses*

Electrical discharge machining occurs at the value of $V_g$, approximately half of the value of operational voltage $V_0$. Idle pulses of a particular amplitude and frequency were detected with the value of the interelectrode gap $\Delta$ (distance between two electrodes) more than the value of the gap of dielectric breakdown $\Delta_{DB}$ ($\Delta > \Delta_{DB}$), Figure 3a. The idle pulse repetition frequency is $f = 10$ kHz,

and the amplitude depends entirely on the factor of the voltage in the interelectrode gap ($V_0$) applied to the electrodes when the electrodes are at a distance Δ.

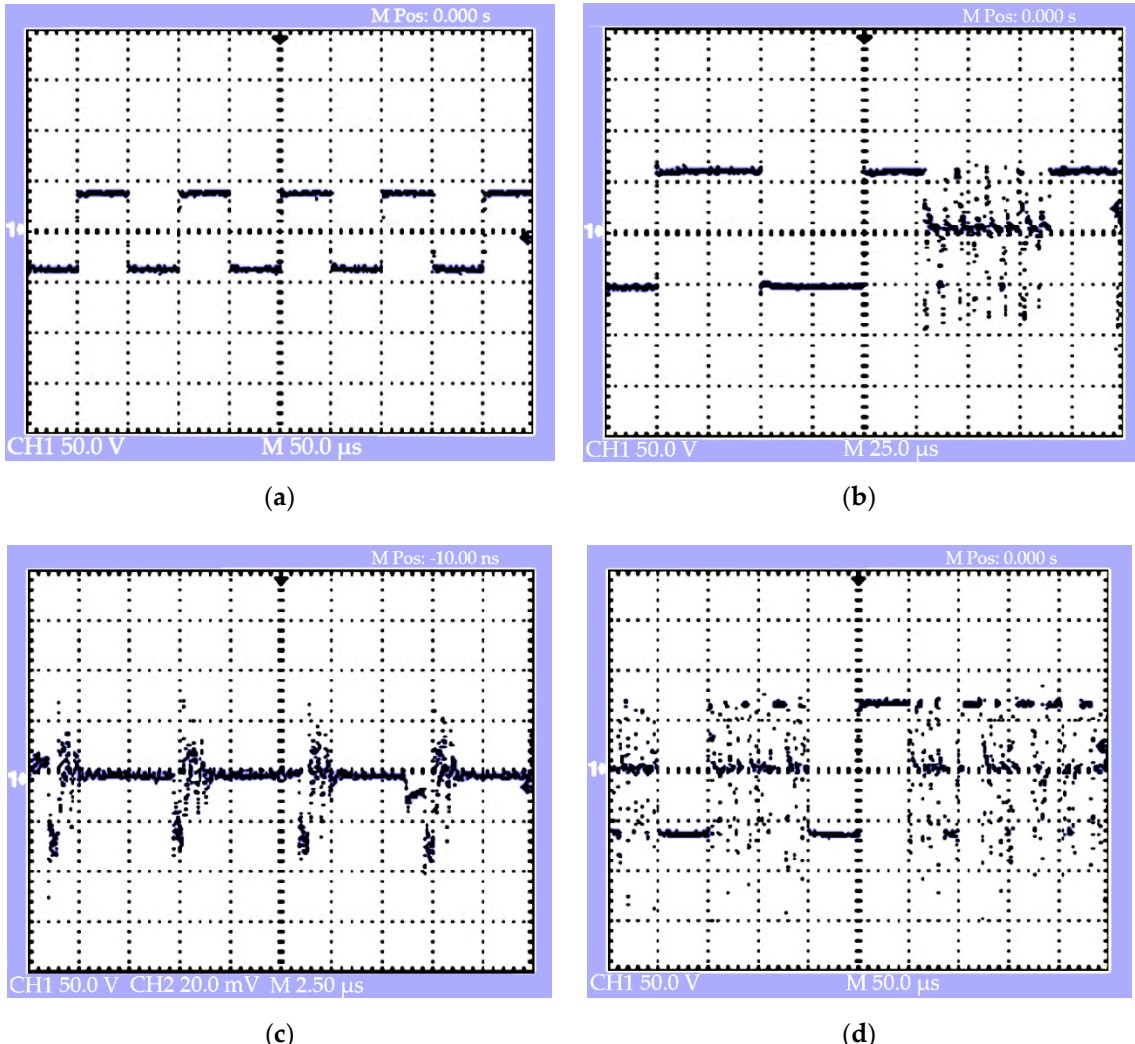

(**a**)

(**b**)

(**c**)

(**d**)

**Figure 3.** Electrical discharge machining pulses detected by the oscilloscope: (**a**) idle pules at a distance Δ > $Δ_{DB}$; (**b**) interrupted introduction of operational pulses at the distance Δ ≥ $Δ_{DB}$; (**c**) operational pulses during machining Δ = $Δ_{DB}$; (**d**) interrupted operational pulses during unstable wire feed and variation with the factors, Δ ≈ $Δ_{DB}$.

An infrequent dielectric breakdown of the gap occurs with a decrease in distance Δ ≥ $Δ_{DB}$ by a single series of pointed operational pulses (Figure 3b). Regular operational pulses that are similar to damped harmonic oscillations followed the tool penetration (Figure 3c). The working pulses frequency is quite high and is equal to tenths of MHz and detected at a level of about 0.2 MHz. Furthermore, the idle and operational pulses start to alternate with different frequencies and amplitudes that depend on factors present in the stable mode (Figure 3d).

### 3.2. Wire Electrode Oscilations

It was determined that the vibroacoustic signal has a periodic character and decreases gradually during the first 2–3 s after tool penetration. It continues to decline during the next 15–20 s of machining (Figure 4a). At the same time, it increases approximately 20 s before the end of the machining. At 5–7 s before the end of machining, the signal interrupts repeatedly. The character of the signal at the beginning and the end of machining has a parabolic character. It corresponds to the final bridge

destruction under the cutoff sample's weight (Figure 4b) that was detected for the samples of both materials with a weight up to 1.8 g.

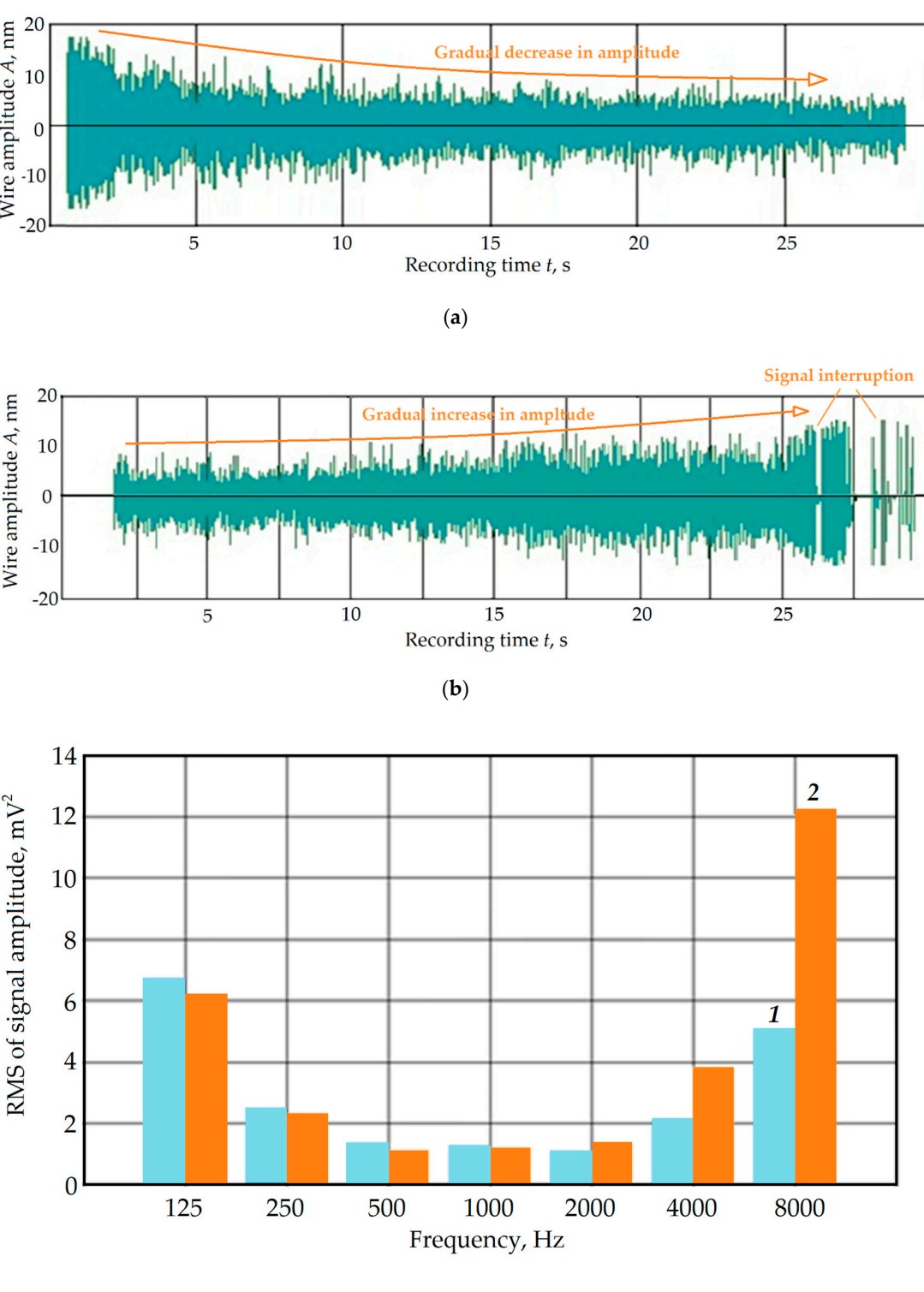

(a)

(b)

(c)

**Figure 4.** Recorded signal and its root-mean-square value during electrical discharge machining of 12Kh18N10T (AISI 321) samples: (**a**) at 30 s of the tool penetration into the workpiece; (**b**) at 30 s before the end of the processing; (**c**) octave spectra of the root-mean-square value of the signal amplitude of 24.5 g sample, $V_0 = 65$ c.u., $W_t = 35$ c.u., where (**1**) at 30 s before the end of the operation, (**2**) at 5 s.

Octave spectra of the root-mean-square value of the signal amplitude (RMS) in Figure 4c showed that RMS differs more than 2.5 times at 60 s and 5 s at a frequency band of 4 ÷ 8 kHz. Simultaneously, the frequency band of 0.125 ÷ 4 kHz does not show significant changes during the recorded periods. The changes in RMS were also observed during the variation of EDM factors and unstable processing.

The changes in a signal's amplitude in the frequency band of 4–8 kHz at 60 s and 5 s before the end of processing has a character that mainly increases with the augmentation of operational factors and weight of the cutting-off sample (Figure 5). The changes at RMS of the signal amplitude at a frequency of 8 kHz were noticeable for 12Kh18N10T (AISI 321) at three various values of the operational voltage Vo. However, the changes in average RMS for the samples of ~2 g from D16 (AA2024) alloy are controversial.

The samples' weight was 24.51 ± 0.0327 g and 4.28 ± 0.0450 g for steel and 10.70 ± 0.0375 g and 1.82 ± 0.1800 g for aluminum for a width of 10 and 2 mm correspondingly based on data of 15 steel samples and five aluminum samples of each width.

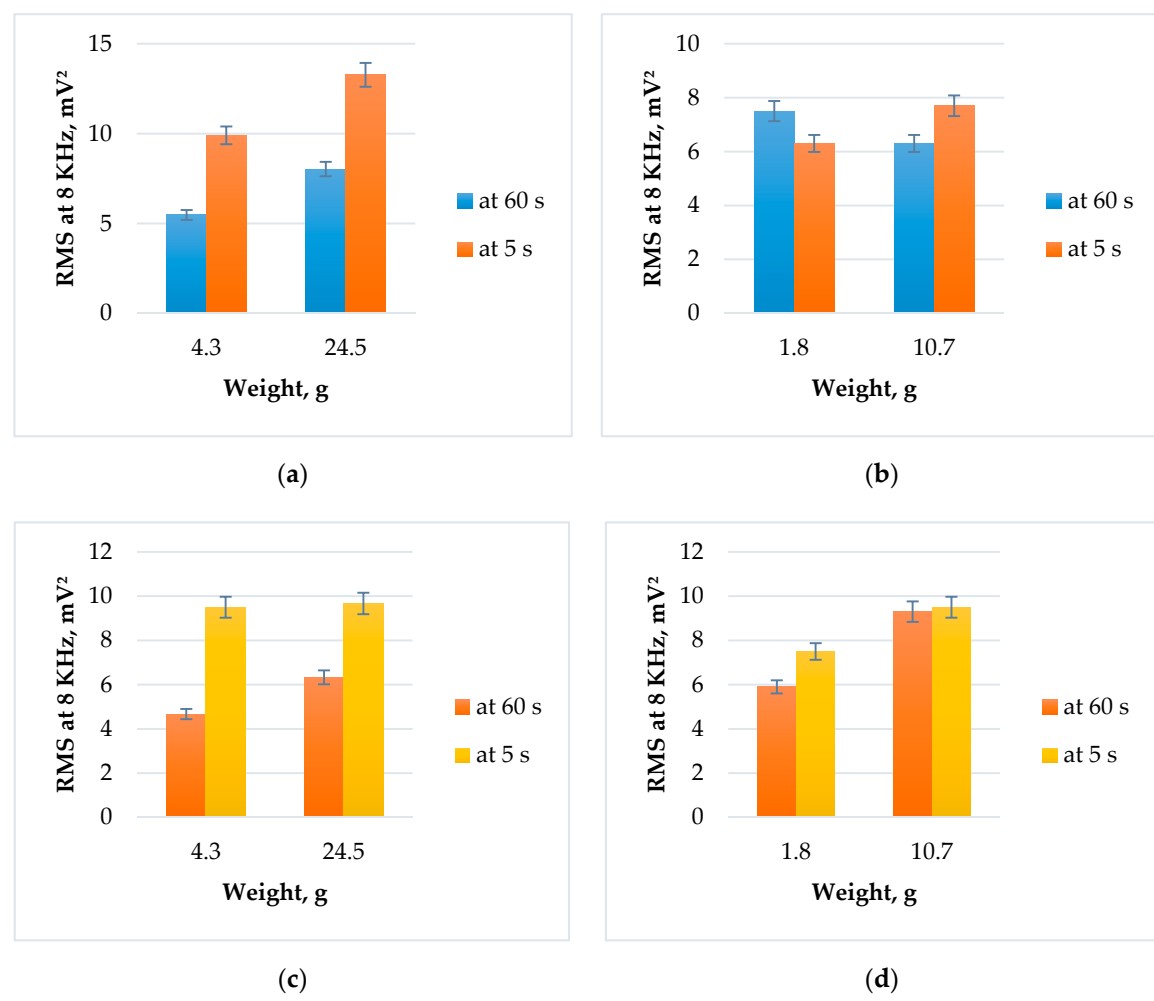

**Figure 5.** *Cont.*

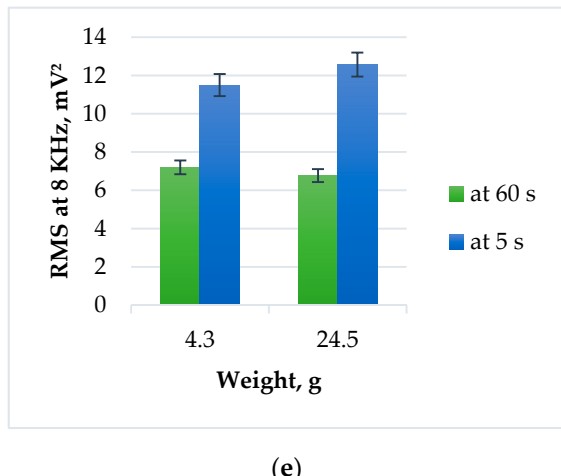

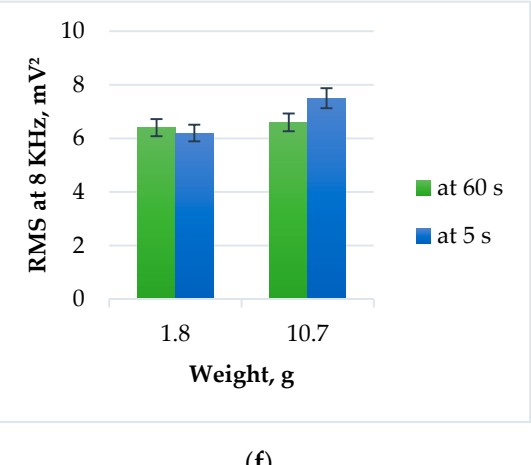

(**e**)                                    (**f**)

**Figure 5.** RMS of the recorded signal's amplitude at 8 kHz for two types of materials at various operational voltage $V_0$: (**a**) 12Kh18N10T (AISI 321) steel, $V_0 = 55$ V; (**b**) D16 (AA2024) alloy, $V_0 = 55$ V; (**c**) 12Kh18N10T (AISI 321) steel, $V_0 = 60$ V; (**d**) D16 (AA2024) alloy, $V_0 = 60$ V; (**e**) 12Kh18N10T (AISI 321) steel, $V_0 = 65$ V; (**f**) D16 (AA2024) alloy, $V_0 = 65$ V.

The changes in RMS of the signal at 60 s and 5 s before the end of processing with the variation in operational voltage $V_0$ and wire tension $W_t$ are more noticeable for 12Kh18N10T (AISI 321) steel than for D16 (AA2024) alloy, that is more ductile (Figure 6). Adequate data were obtained even for tiny pieces with a weight of 3.7 g and 1.8 g correspondingly. It should be noted that stable processing corresponds to the RMS's minimal value at 60 s. RMS of the signal amplitude is higher at 12Kh18N10T (AISI 321) steel machining (Figure 6a,b), by 12.5%, compared to D16 (AA2024) alloy machining (Figure 6c,d). RMS of steel is in the range of $5 \div 14$ mV$^2$ with arithmetic mean of 8.54 mV$^2$; RMS of duralumin is in the range of $4.5 \div 9$ mV$^2$ with arithmetic mean of 7.475 mV$^2$.

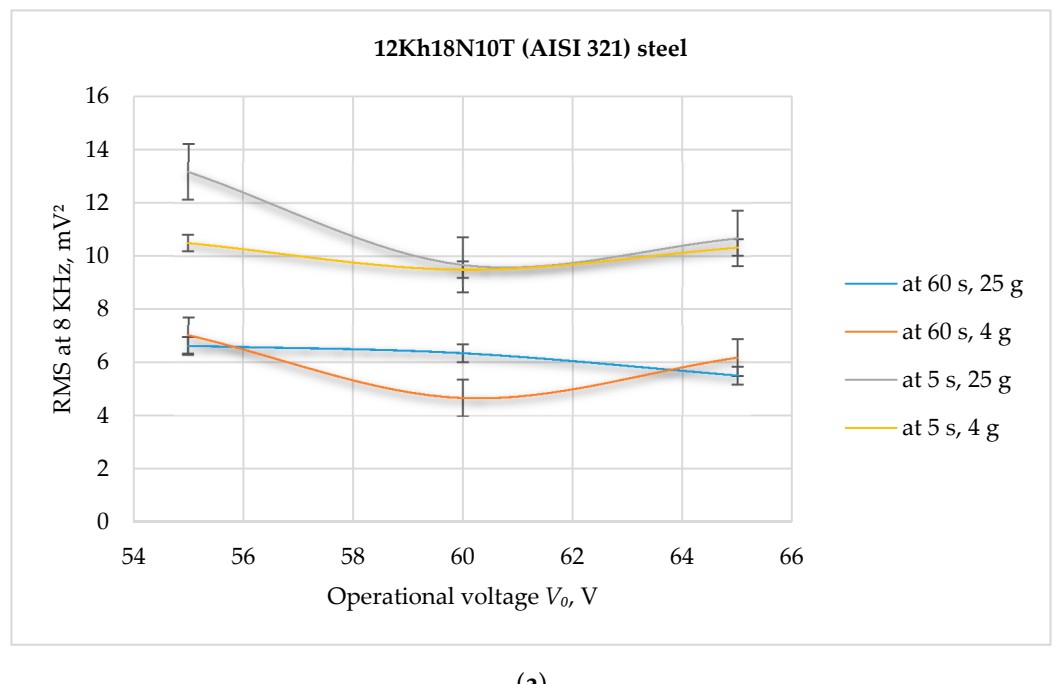

(**a**)

**Figure 6.** *Cont.*

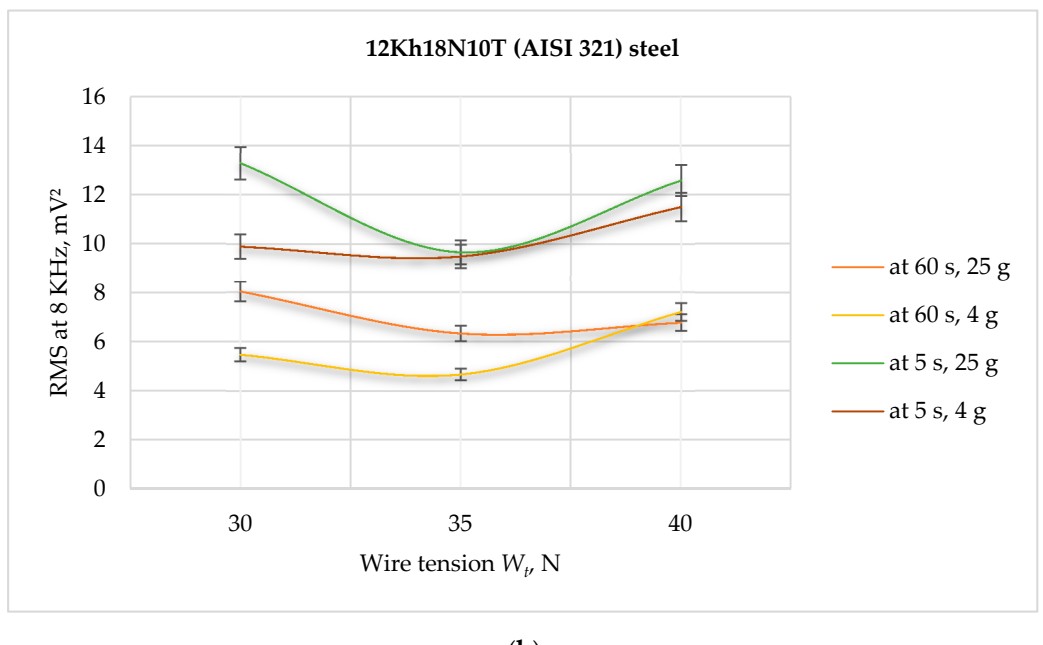

(**b**)

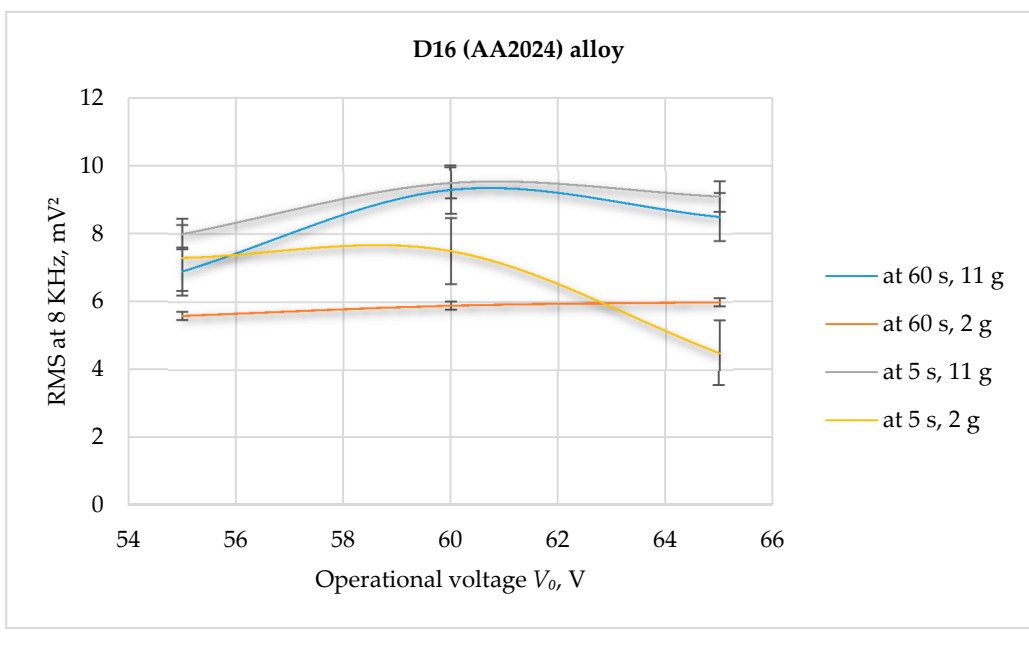

(**c**)

**Figure 6.** *Cont.*

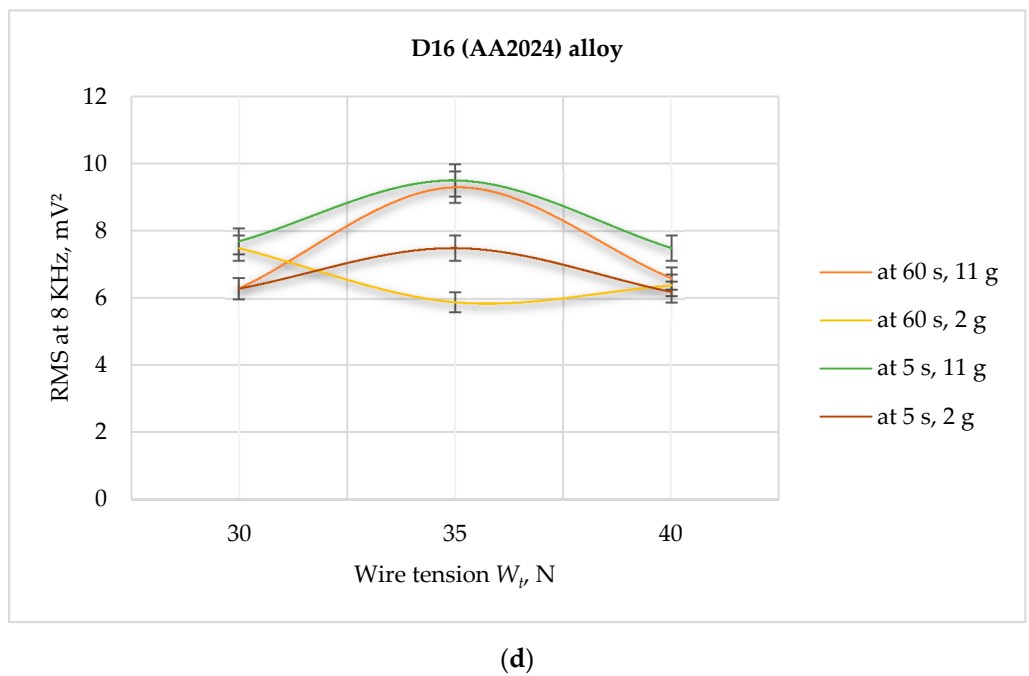

(**d**)

**Figure 6.** RMS of the received electrical discharge machining oscillation signal at 8 kHz for two types of materials: (**a**) 12Kh18N10T (AISI 321), in dependence on operational voltage $V_0$; (**b**) 12Kh18N10T (AISI 321), in dependence on wire tension $W_t$; (**c**) D16 (AA2024), in dependence on operational voltage $V_0$; (**d**) D16 (AA2024), in dependence on wire tension $W_t$.

### 3.3. Morphology of the Samples

Roughness profile *Ra* and recorded signal at 30 s before the end of machining are presented in Figure 7. As can be seen, the density of the signal amplitude is higher at 12Kh18N10T (AISI 321) steel machining (Figure 7a,b), by 20%, compared to D16 (AA2024) alloy machining (Figure 7c,d); approximately 30 $\mu m^{-1}$ and ~24 $\mu m^{-1}$, correspondingly. The three-dimensional (3D) graphs of the EDM factors' influence on the average roughness *Ra* are presented in Figure 8, where minimal value is associated with the stable machining process and the lowest RMS values of the signal's amplitude.

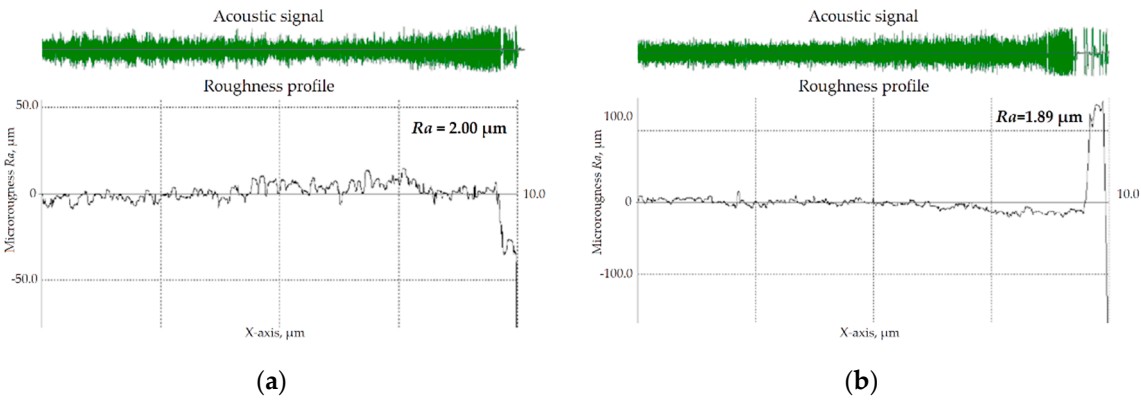

(**a**)　　　　　　　　　　　　　　　　　　　　　　(**b**)

**Figure 7.** *Cont.*

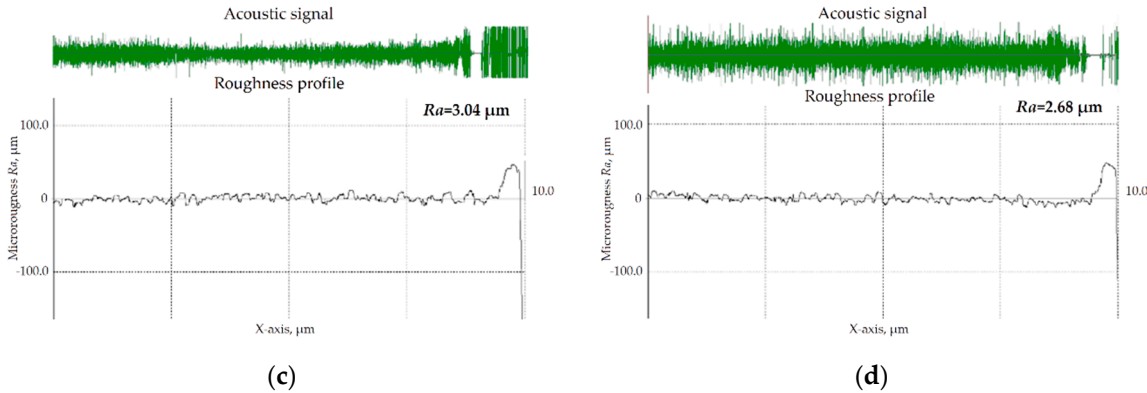

**Figure 7.** Roughness $R_a$ and recorded signal amplitude: (**a**) 12Kh18N10T (AISI 321) steel of 24.5 g, $R_a$ of 2.00 μm, $V_0 = 55$ V; $W_t = 35$ N; (**b**) 12Kh18N10T (AISI 321) steel of 4.3 g, $R_a$ of 1.89 μm, $V_0 = 55$ V; $W_t = 35$ N; (**c**) D16 (AA2024) alloy of 10.7 g, $R_a$ of 3.04 μm, $V_0 = 60$ V; $W_t = 35$ N; (**d**) D16 (AA2024) alloy of 1.8 g, $R_a$ of 2.68 μm, $V_0 = 60$ V; $W_t = 40$ N.

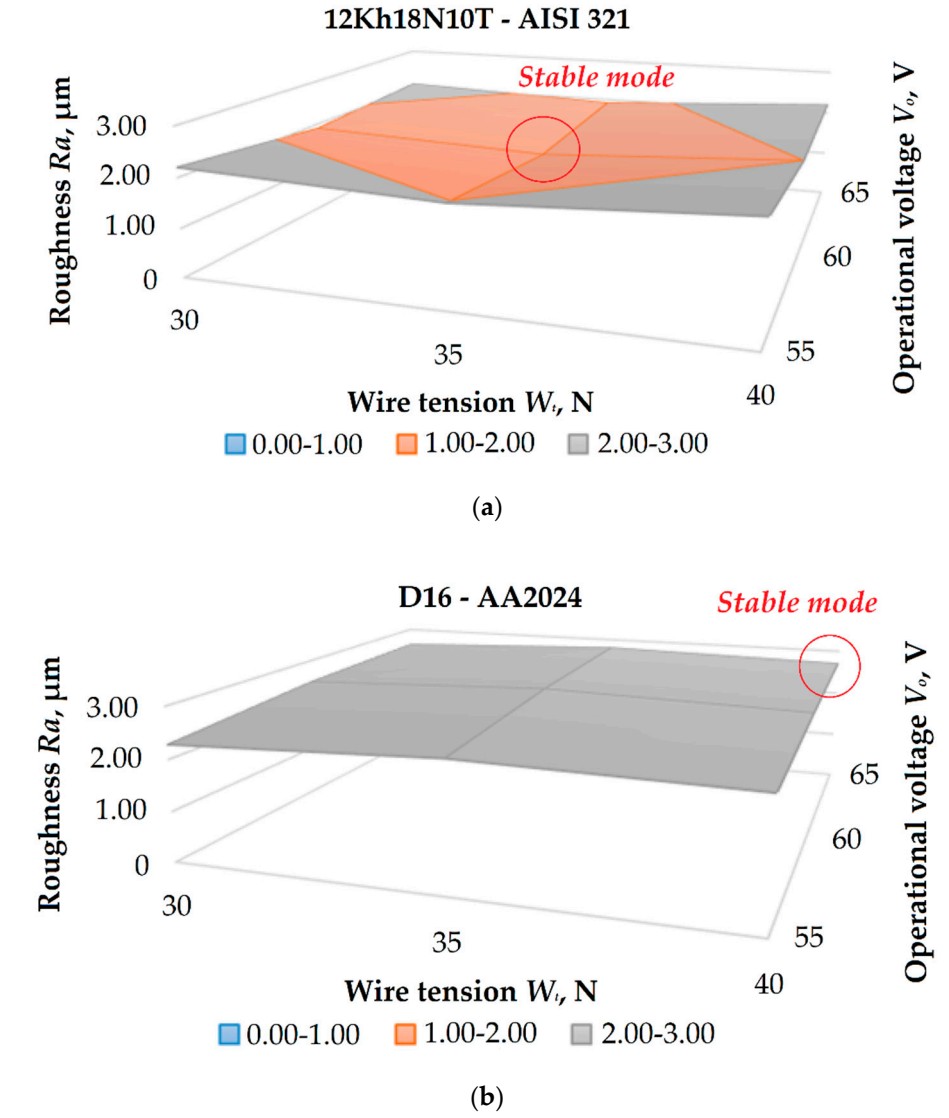

**Figure 8.** Three-dimensional (3D) graphs of the surface roughness $R_a$ dependences on operational voltage $V_0$ and wire tension $W_t$: (**a**) 12Kh18N10T (AISI 321) steel; (**b**) D16 (AA2024) alloy.

### 3.4. Discharge Gap

Figure 9 shows the measured offset $\Delta^*_{DB}$ of the path in the dependence of EDM factors for two types of materials. The offset $\Delta^*_{DB}$ includes the wire radius $r_w$ of 0.125 mm. The optically measured effective discharge gap $\Delta_{DB}$ is in the range of $45 \div 53$ μm for 12Kh18N10T (AISI 321) steel and in the range of $71 \div 78$ μm for D16 (AA2024) alloy. The minimal values—170 μm for anti-corrosion steel and 196 μm for aluminum alloy—are associated with the stable machining process and corresponds to the lowest RMS values of the signal's amplitude.

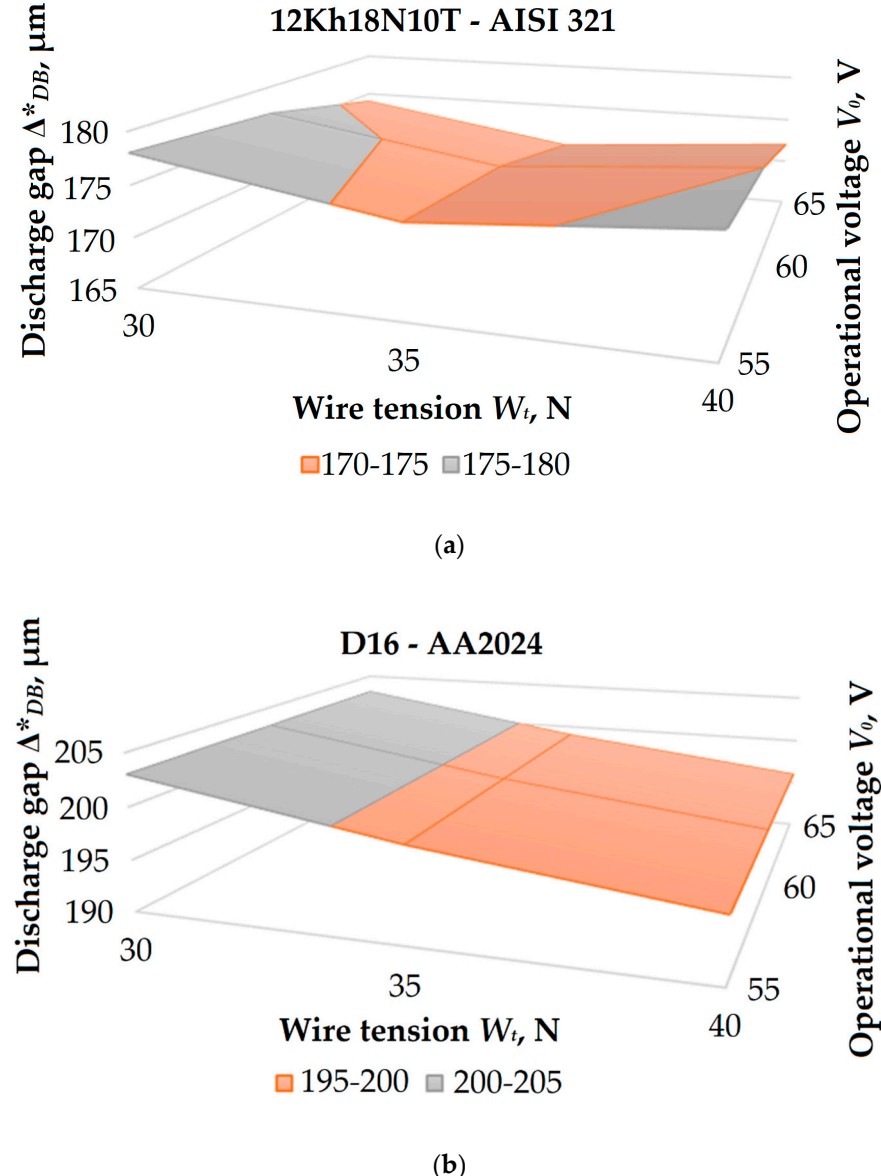

**Figure 9.** 3D graphs of the offsets $\Delta^*_{DB}$ dependencies on operational voltage $V_0$ and wire tension $W_t$: (**a**) 12Kh18N10T (AISI 321) steel; (**b**) D16 (AA2024) alloy.

### 3.5. Tool Wear

The tool electrode's rupture point (Figure 10) shows cup neck formation before destruction that corresponds to the ductile properties of the brass with the reduction area:

$$S_{RA} = \frac{0.049 - 0.003}{0.049} \cdot 100 = 93.8 \, [\%], \tag{13}$$

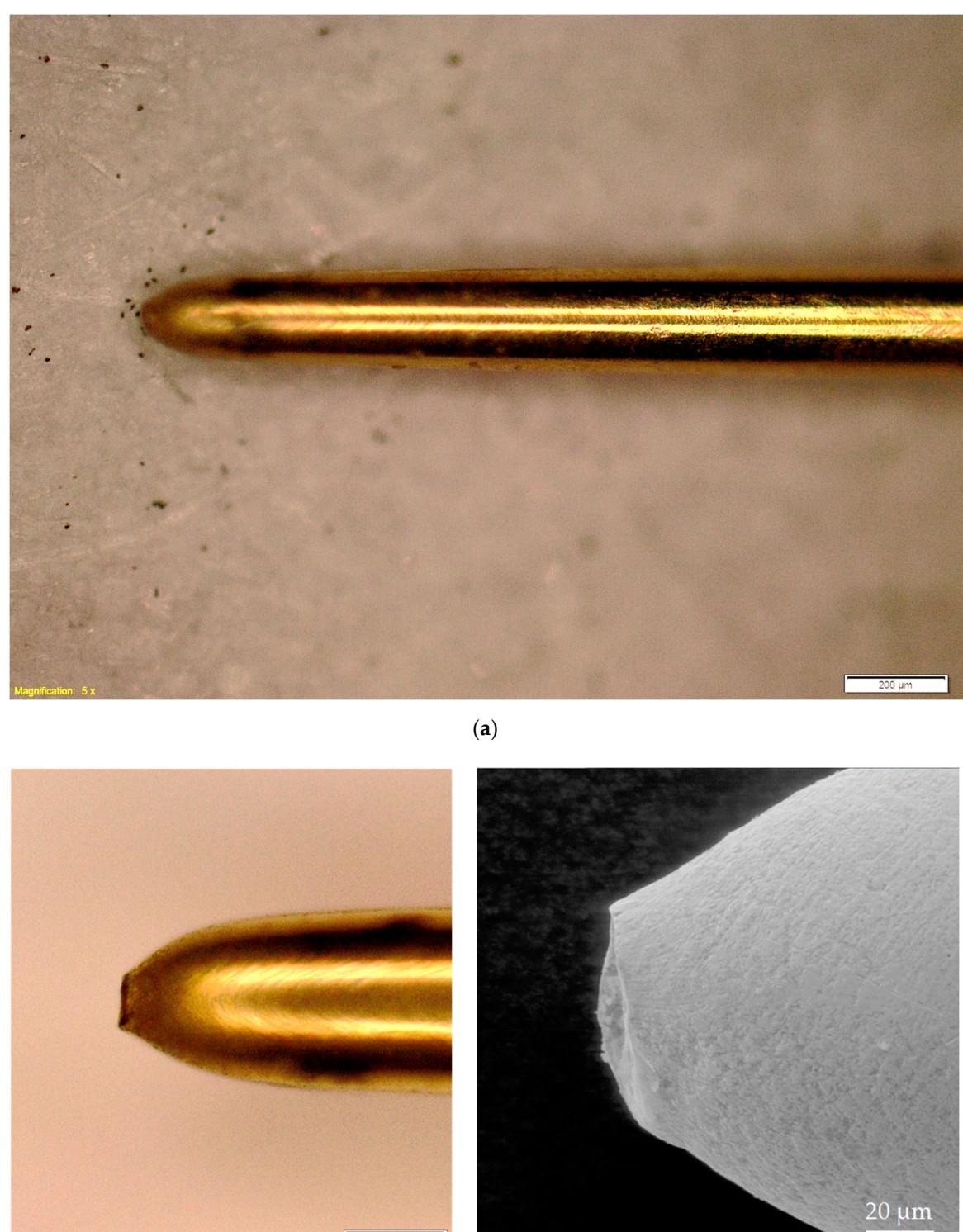

**Figure 10.** Wire tool: (**a**) microphotograph, 5×; (**b**) microphotograph, 10×; (**c**) SEM-image, 2.0k×.

Figure 11 presents the electrode wear after roughing and finishing at the electrical discharge machining of 12Kh18N10T (AISI 321) steel and D16 (AA 2024) duralumin.

Figure 11a shows the conjugation of two interdependent surfaces with two types of wear—lateral at left and front at right. The wear has a different character. The front wear surface has the appearance of the typical eroded surface—sublimated and heat-affected material coated by the secondary structure

of electrode components with pores and cracks. Moreover, the surface is covered by craters of solidified secondary material—usually consisting of the metallic material of the first order, solid solutions, and complex compounds of the second order (mostly oxides in the case of machining in deionized water). The craters have an explosive character that is not observed at lateral surfaces. The line of two wear surfaces conjugation is pronounced.

Figure 11b shows lateral wear at finishing. The formed surface has visible edges; the conjoined surface's left side has no presence of wear when the right side is also blank but with clear traces of secondary structure explosive deposition at the blank surface. The lateral wear surface showed two types of material destruction—the classical eroded surface of material sublimation with secondary structure deposition and mechanical wear traces.

Figure 11c shows the conjugation of two surfaces—of lateral wear and blank surface at roughing. The left side of the image—blank surface—has pronounced traces—drops, copious splashes—of explosive character of interaction occurred in the discharge gap at lateral wear. A significant volume of uneven sublimated material coated by the secondary structure with pores and cracks presents the surface with lateral wear at the right side of the image.

The front wear surface at roughing (Figure 11d) has secondary structure pellet formation that coat the sublimated surface. The secondary sublimated surface shows typical nanoframe structure formation—more easy-to-melt material components sublimate from the secondary structure's coating (pellets) and are adsorbed by the refractory matrix.

Figure 11e shows the conjugation of two surfaces—front and lateral wear at roughing. The left side of the image—front wear surface—has a coating of secondary structure.

Figure 11f shows the lateral wear's surface at finishing when the obtained surface has traces of two types of wear—thermal and mechanical, which can be easily identified.

The cross-sections of the electrodes at roughing and finishing are presented in Figure 11g,h. Both the cross-sections showed quite intensive wear with affluent loose of the electrode material. The worn area, volume, and mass of the tool, volumetric, and mass wear rates were calculated using Equations (8)–(10) (Tables 6 and 7).

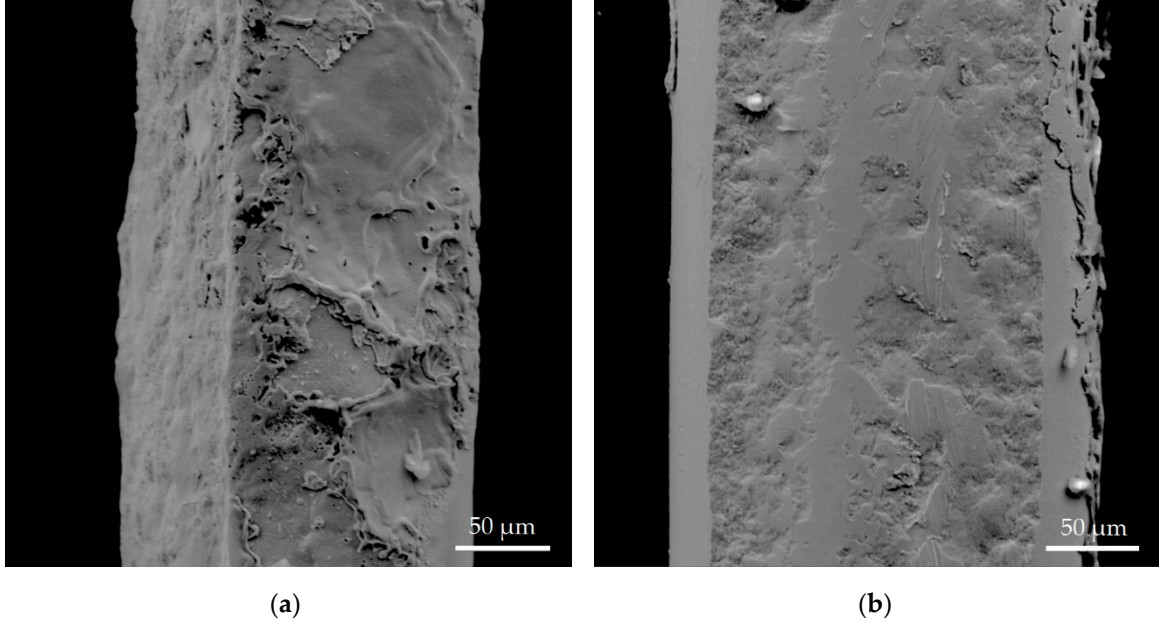

(**a**)                     (**b**)

**Figure 11.** *Cont.*

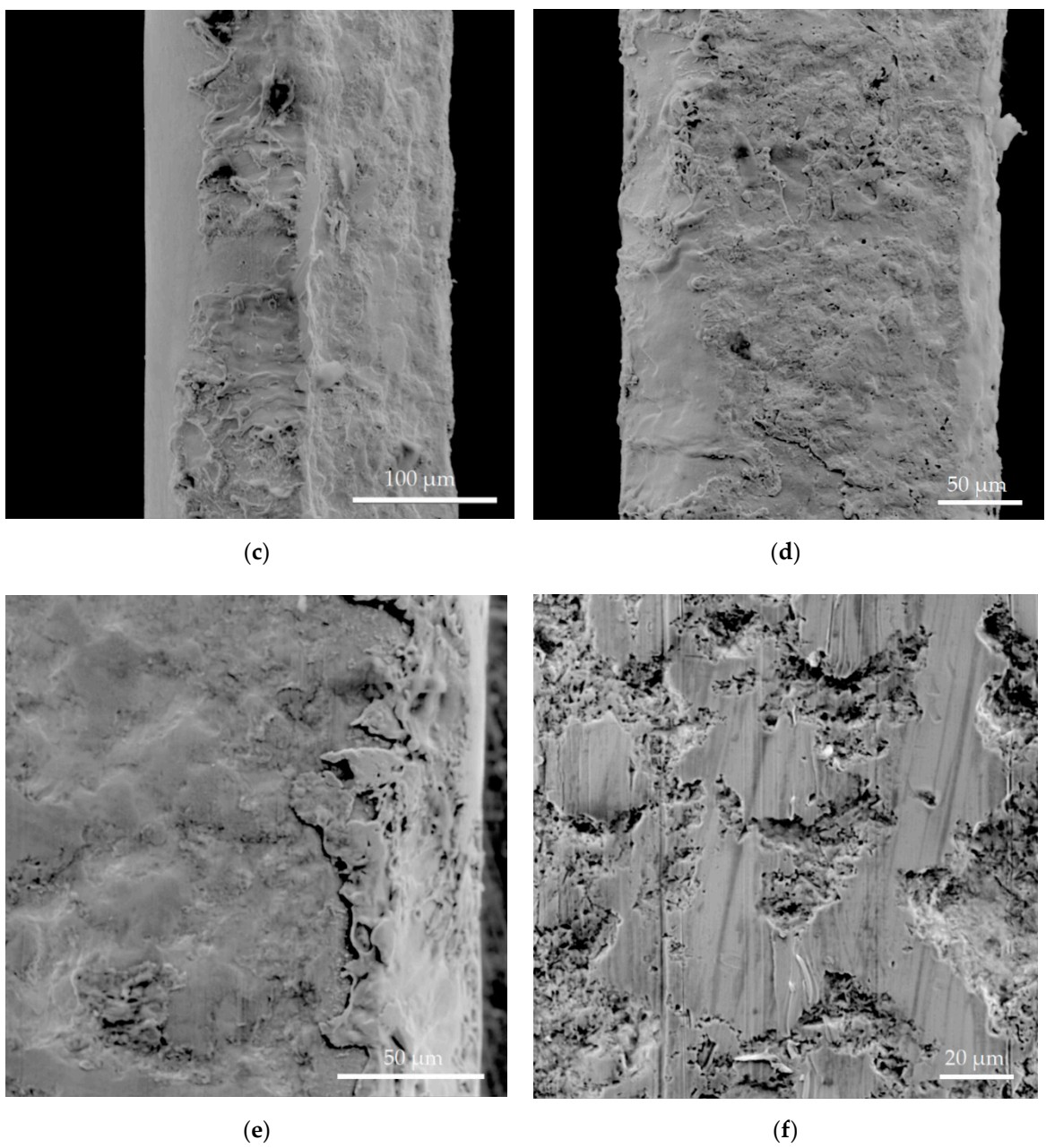

(**c**)　　　　　　　　　　　　　　　　　　　　(**d**)

(**e**)　　　　　　　　　　　　　　　　　　　　(**f**)

**Figure 11.** *Cont.*

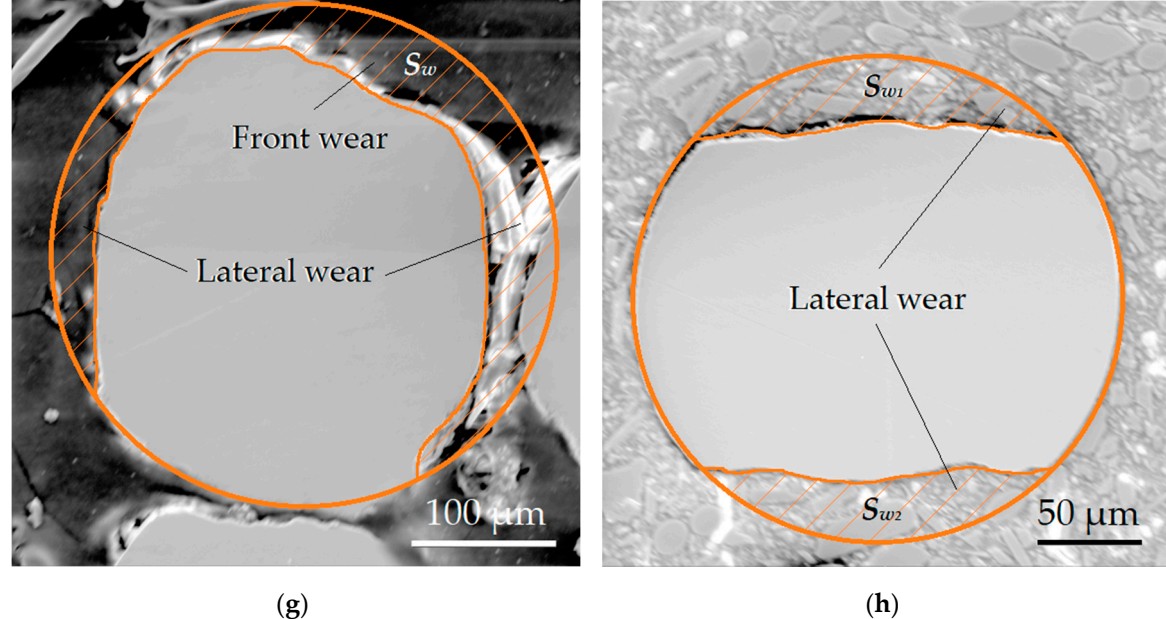

(**g**)                                    (**h**)

**Figure 11.** SEM-images of wire tool electrode sample: (**a**) conjugation of front and lateral wear at roughing of 12Kh18N10T (AISI 321) steel, 916×; (**b**) lateral wear at finishing of 12Kh18N10T (AISI 321) steel, 909×; (**c**) conjugation of lateral wear and blank surface at roughing of 12Kh18N10T (AISI 321) steel, 780×; (**d**) front wear at roughing of D16 (AA2024) duralumin, 909×; (**e**) conjugation of front and lateral wear at roughing of D16 (AA2024) duralumin, 1.61k×; (**f**) lateral wear at finishing of D16 (AA2024) duralumin, 2.04k×; (**g**) cross-section after steel roughing, 696×; (**h**) cross-section after steel finishing, 1.02k×.

**Table 6.** Volumetric wear rate of electrical discharge machining of 12Kh18N10T (AISI 321) steel.

| Type of Machining | Worn Surfaces | Measuring Error | Worn Area $S_w$ | Summarized Worn Area $S_s$ | | Volumetric Wear $\Delta V$ | Volumetric Wear Rate [2] $R_v$ | Error |
|---|---|---|---|---|---|---|---|---|
| | | [μm] | [mm²] | [mm²] | [%] [1] | [mm³] | [mm³·s⁻¹] | [μm³·μs⁻¹] |
| Roughing | Front | ± 1 ÷ 2 | 0.0014 ± 0.00015 | 0.021 ± 0.0022 | 51 ± 10.5 | 0.42 ± 0.0002 | 1.22 ± 0.04 | ± 20 ÷ 40 |
| | Lateral | | 0.0048 ± 0.0010 | | | | | |
| Finishing | Lateral | | 0.0045 ± 0.00005 | 0.009 ± 0.0001 | 18 ± 5.56 | 0.18 ± 0.00001 | 0.52 ± 0.002 | |

[1] Calculated to the entire cross-sectional area; [2] rewinding rate of 3.5 m/min.

**Table 7.** Mass wear rate of electrical discharge machining of 12Kh18N10T (AISI 321) steel.

| Type of Machining | Measuring Error | Worn Mass [1] $\Delta m$ | Mass Wear Rate [2] $R_m$ | Error |
|---|---|---|---|---|
| | [g] | [g] | [g·s⁻¹] | [g·μs⁻¹] |
| Roughing | ±0.0001 | $3.3 \times 10^{-3} \pm 0.00005$ | $9.6 \times 10^{-3} \pm 0.01$ | ± 0.01 ÷ 0.02 |
| Finishing | | $1.4 \times 10^{-3} \pm 0.00004$ | $4.0 \times 10^{-3} \pm 0.008$ | |

[1] Density of $7.9 \times 10^3$ kg/m³ or 0.0079 g/mm³ at +20 °C; [2] rewinding rate of 3.5 m/min.

### 3.6. Chemical Content

The chemical content of the tool electrode's cross-section at roughing of 12Kh18N10T (AISI 321) steel is presented in Figure 12.

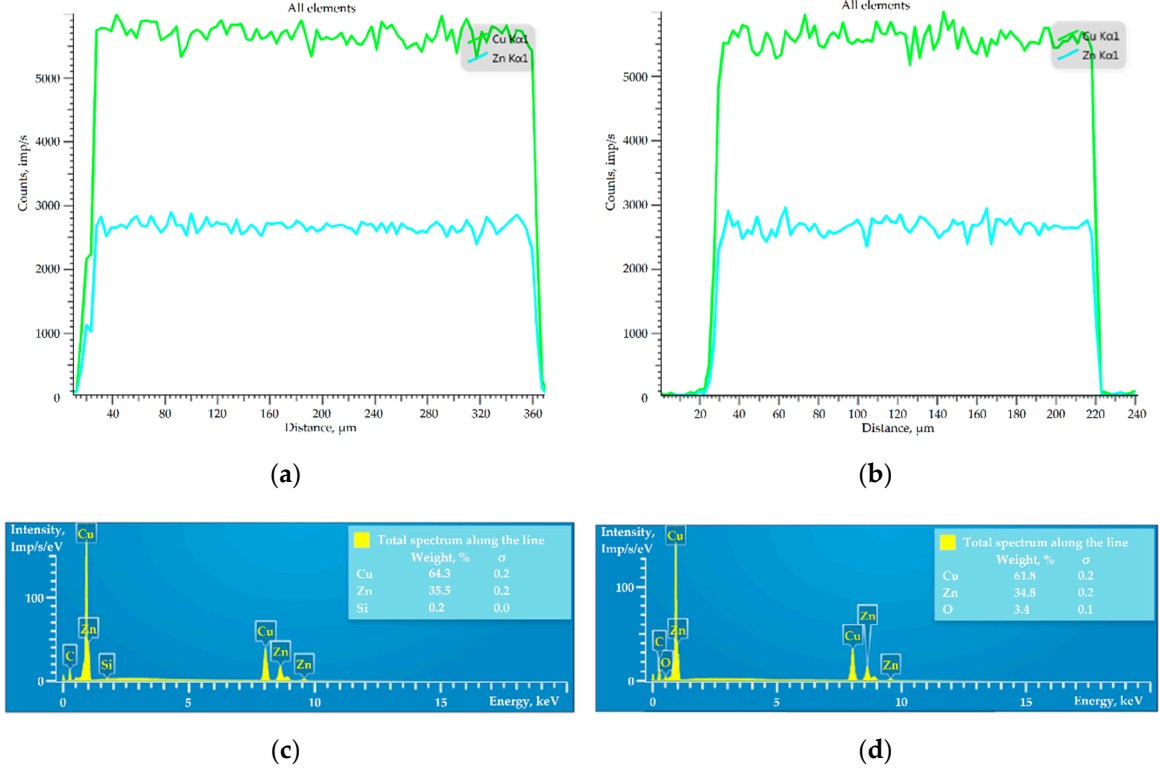

**Figure 12.** Chemical analyses of the worn tool electrode cross-section after machining: (**a**) chemical elements along the line at front wear; (**b**) chemical elements along the line at lateral wear; (**c**) EDX spectrum at front wear; (**d**) EDX spectrum at lateral wear.

## 4. Discussion

### 4.1. Discharge Pulses and Oscillations Control

Currently, a large number of EDM factors determine the machining mode, the value of which adaptively adjusts by the CNC system during processing. It is called an adaptive pulse-width modulation based on electrical response (Figure 3). At the same time, the value of interelectrode gap, the stability of processing, and consequently the quality of the obtained surfaces depend on the homogeneity of workpiece structure and microstructure, effectiveness of the erosion debris washout by the working fluid, the workpiece thickness, and electrophysical and electrochemical properties of the materials in the working zone. In this connection, the discharge pulses have a more chaotic, probabilistic nature, depending on many factors.

The vibroacoustic signal reflects the changes in the weight and structure of the workpiece, the main discharging factors that correlated with the force diagram in the working zone [27,55] that influence the amplitude of the signal in the wide range of spectra [23,25,26,51,52].

The recorded signal arises during processing and increases by 1.5 times from the initial level at the end of processing (Figure 4). The signal interruptions can be observed at 5 s before the end of processing, which is associated with the direct contact between the workpiece and tool electrode that occurs during changes in the cutting-off sample position in the working space in relation to the rest of the workpiece. It leads to the consequent clamping of the tool electrode to the workpiece by moving the sample and short circuits.

Optimum electrical discharge machining factors have the least value compared to the closest values (Figure 6a–c, except for the duralumin of 2 g). The EDM factors for the stable electrical discharge machining are $V_0 = 60$ V, $W_t = 35$ N for steel and $V_0 = 55$ V, $W_t = 30$ N for duralumin. At the same time, the sensibility of the system grows with its stiffness (Figure 6b,d) and decreases with the weight of the sample (Figure 6c,d).

An increase in RMS of the signal amplitude for stable factors at 8 kHz at 5 s before the end of processing was $40 \div 55\%$ for steel and $12.5 \div 25\%$ for aluminum alloy compared with data recorded at 60 s (Figure 5b,c and Figure 6).

The developed system showed its controversial response for the samples of 2 g of aluminum and adequate data for the samples' weight more than 4 g for the steel and 10 g of aluminum. The samples' weight varies by more than six times, but the RMS of the recorded signal demonstrates similar trends.

The observed behavior of the signal (Figures 5 and 6) can be correlated with the particular features of elastic and plastic deformation during the ductile failure according to the stress-strain curves and scheme of fracture formation (Figure 2) as aluminum shows better ductility during destruction that actually associated with a stretch of the interatomic bonds [56–60]. It correlates with the recorded signal when more brittle material—steel shows an adequate signal response even for the samples of 4 g when data received for ductile aluminum alloy are less significant but can also be registered for monitoring and control of samples of more than 2 g. That all make a basis for the development of multi-parameter control systems and switch to the next technological paradigm [61–67].

The minimum value of the measured roughness $R_a$ of the samples (Figure 8) correlates with stable machining signals. The same tendency is observed for the measured discharge gap (Figure 9).

### 4.2. Wire Breakage and Tool Wear

The optical and scanning electron microscopy (Figure 10) showed that the character of rupture had mechanical nature corresponding to cup neck formation and stress-strain curve of middle ductile material destruction—brass alloy [47–49]. The observed area has the topology of the wire breakage that occurs in most of the cases during electrical discharge machining with unstable factors, in case the surface inclination or uneven structure of the workpiece need to be processed. There is no presence of thermal defects except an ashy shade at the formed cup. Additionally, there is no evidence of the rupture's external origin that can occur during the wire cut.

The excess in bias during wire blockage between the workpiece and cutting-off sample that did not allow adequate debris removal probably caused this rupture, since the current and pulses factors were constant. An increase in bias gave a denser distribution of discharges, while an increase in current gave more expressed discharges [28–30] (Figure 13a).

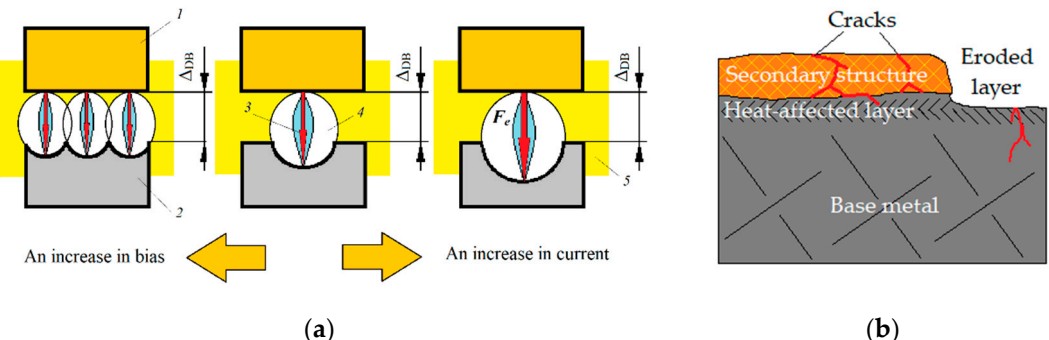

**Figure 13.** Electrical discharge machining principles: (**a**) dependencies of the discharge pulse character on bias and current, where (**1**) is a tool electrode, (**2**) is a workpiece, (**3**) is a discharge channel, (**4**) is a plasma cloud, (**5**) is dielectric medium, and $F_e$ is discharge force; (**b**) submicron structure of erosion wear.

The calculated enlarged value of the reduction area ($S_{RA} = 93.8\%$) is obviously caused not only by mechanical rupture but a mechanical rupture in the softened state [68,69] due to the heat of the discharge gap that was definitely above 600–650 °C (dark red color) since the brass's surface around the formed cup neck is covered by the ashy shade of zinc oxide (Figure 10a) [70–72].

The formed craters that have different from the typical erosion morphology are 30–100 μm (Figure 11a). The explosive droplets reached a distance of ~100 μm from the wear edge on the backside

surface (Figure 11c). The submicron structure of these droplets (Figure 11e) is different from the typical eroded surfaces as on the lateral surface at roughing and at finishing (Figure 11a,b,e,f).

The observation area of the wire tool presented in the SEM-microphotographs showed:

- Thermal traces (Figure 11a,e), which partly have a relation to the erosion that occurred between electrodes;
- Mechanical traces (Figure 11b,f), which has no relation to the processing;
- A topology that is more correlated to the electrical erosion of the materials under discharge pulses (Figure 11c,d).

The samples with typical erosion wear traces (Figure 11c,d) correlate to non-oxide (oxygen unsaturated) structures [73,74]—secondary submicrostructures of the complex compounds (of second order) adsorbed by the eroded surface of the base material—of the first order (Figure 14b) [75–77], which probably contain metastable and insoluble solid solution in the form of adherent and brittle thin film and heat-affected sublayer [78–81].

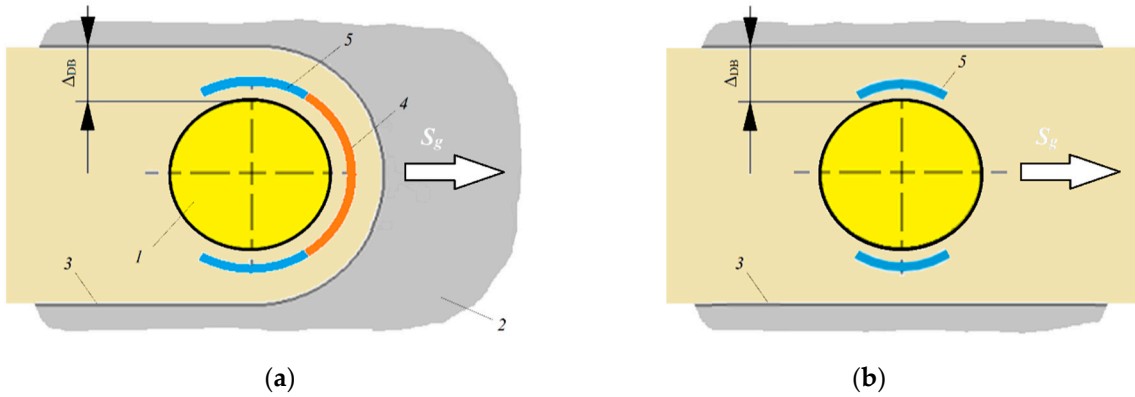

               (**a**)                                       (**b**)

**Figure 14.** Non-profiled tool electrode wear: (**a**) at roughing; (**b**) at finishing, where (**1**) is a tool electrode, (**2**) is a workpiece, (**3**) is machined surface, (**4**) is front wear, (**5**) is lateral wear, and $S_g$ is a guidance feed of wire.

The mechanical traces on the surface of the wire at lateral wear correspond to the mechanical destruction of the tool during rewinding (Figure 11b,f) that occurred after electrical erosion (secondary wear). Probably, wire tool pinch rollers or diamond nozzle destroyed the morphology of the lateral wear mechanically.

The thermal traces were very pronounced at roughening (front wear), which are different from the lateral wear morphology that was especially apparent at the conjunction of the front and lateral wear surfaces (Figure 11a,e), and the lateral and blank surfaces (Figure 11c) have a different origin, related to the chemical composition of the tool and workpiece.

This is due to the picture of the erosion process observed visually and based on the interaction of the components of the electrodes—CuZn35 brass alloy and 12Kh18N10T (AISI 321) steel. The nickel and zinc reaction at a temperature of 1000 °C has an explosive character and results in the formation of intermetallic $ZnNi_x$ (x = 0, 5, 10, 15, 20 wt%) [82–86]. It can be easily observed by the formation of non-periodic orange flashes in the discharge gap with the release of abundant black sediment during the processing of anti-corrosion austenite (nickel-containing) steels [25,34,87–89]. Visually, the density of the flashes is less than the density of discharges and occurs with a lower frequency. However, the flashes cannot be detected with a higher workpiece, especially with a height of more than 100 mm that often occurred at machining in tool and mold production, when the visual access to the working zone is absent. Thus, the signal was adequately registered by the developed vibroacoustic diagnostic mean—RMS of the amplitude signal was higher by 12.5% and more intense by 20% for 12Kh18N10T (AISI 321) steel than for D16 (AA 2024) alloy for the thickness of the sample of 20 mm.

As can be seen, the deposition of the secondary submicron structure of the sublimated electrodes' components and working medium in the case of anti-corrosion steel processing occurred explosively (with craters of 30 ÷ 100 µm). The presence of the explosive character of reaction between metals accompanies the electrical erosion wear can be seen in Figure 11c on the blank surface when a clear edge limits the area of deep EDM wear on the lateral surface during roughing. However, the explosive nature of the secondary phase deposition overcomes the wear edge and is visible from the electrode's blank side.

The front wear at roughing has a more pronounced topology that correlates with the non-oxidized erosion wear an explosive reaction between wire and workpiece components (Figure 14), where the deposed film of secondary structure coat eroded base metal surface. The lateral wear at roughing has a less pronounced topology that corresponds to the typical wear that occurred under discharge pulses. It correlated to the degree of the involvement of the sides of the electrode in the formation of the slot when the front surface has the presence of secondary wear of the formed films: the front side is more involved in the formation of the slot, and the side surfaces are involved in the erosion process only partly by secondary "polishing" formed surfaces [24,90,91]. The lateral wear at finishing has a similar character. However, wire tool pinch rollers destroyed the morphology of the lateral wear mechanically.

The electrode's cross-section shows the intensity of the two types of wear (Figure 11g,h). In the considered sample, the front wear does not predominate the lateral one at roughing, and distributes quite even at the periphery of the tool (Figure 11g). The conjugation of the worn surfaces was pronounced for all of the samples.

Analysis of chemical elements along the line and EDX spectrum of the wire tool at roughing and finishing (Figure 12) showed mostly chemical elements except for chemical elements of the brass wire in balance—61.8 ÷ 64.3% of Cu and 34.8 ÷ 35.5% of Zn. However, less than 3.4% of oxygen is proof of semiconductive and amphoteric zinc oxide formation, which usually occurs during brass heating (Figure 10a) [92,93], when copper (II) oxide decomposes in the presence of hydrogen [94,95]:

$$2ZnO + O_2 \rightarrow 2ZnO, \tag{14}$$

$$CuO + H_2 \rightarrow Cu + H_2O. \tag{15}$$

Both of the oxides do not interact with water. Zinc oxide gets yellow with heating and sublimates at 1800 °C. It should be noted that that oxygen was present quantitatively more in the samples after finishing and at later wear of roughing, while it was not possible to quantify it along the line in some cases at front wear after roughing. A small amount of carbon that was not quantitatively evaluated (less than 0.2%) is associated with normal atmospheric contamination.

## 5. Conclusions

### 5.1. Monitoring System and Tool Behavior

A comprehensive study of the tool electrode's wear process during electrical discharge machining was accomplished by the developed monitoring system based on oscillation detecting. That gives detailed data on the character of electrode tool wear and stability of workpiece machining in the high-frequency acoustic band of 8 kHz.

The optimum electrical discharge machining factors are detected by monitoring the vibroacoustic signal—RMS value of the amplitude at 8 kHz for steels and more ductile duralumin with a weight of more than 2 g. The stable electrical discharge machining are $V_0 = 60$ V, $W_t = 35$ N for steel and $V_0 = 55$ V, $W_t = 30$ N for duralumin. An increase in RMS of the signal amplitude at 5 s before the end of processing was 40 ÷ 55% for steel and 12.5 ÷ 25% for aluminum alloy compared with data recorded at 60 s. The proposed approach can be used to develop a multiparameter controlling system of EDM-equipment to carry out the modern CNC-systems at a principally new level.

### 5.2. Wire Tool Topology and Wear Rate

Classification of the obtained surface topology of the tool electrode determines two types of wear under discharge pulses related to the thermal nature: material sublimation and chemical interaction between components of the working zone when mechanical destruction of the finishing electrode sample has a different origin.

Volumetric wear rate $R_v$ was $1.22 \pm 0.04$ mm$^3 \cdot$s$^{-1}$ at roughing and $0.52 \pm 0.002$ mm$^3 \cdot$s$^{-1}$ at finishing; mass wear rate $R_m$—$9.6 \times 10^{-3} \pm 0.01$ g·s$^{-1}$ and $4.0 \times 10^{-3} \pm 0.008$ g·s$^{-1}$, respectively. $41 \div 62\%$ of the tool subjected wear under discharge impulses at roughing during electrical discharge machining of anti-corrosion steel when summarized lateral wear exceed front wear by 29.17%. $12 \div 24\%$ of the tool sublimates under lateral wear at finishing.

The study showed that the processing of the materials with inadequate process parameters or the not proper combination of tool and workpiece materials causes more intensive wear of the tool correlated with the chemical interaction of the electrodes and dielectric medium components. This leads to the micro explosive character of processing with formation intermetallic ZnNix (x = 0, 5, 10, 15, 20 wt%), with Zn of the brass and nickel of austenite steel that was also registered the mean of vibroacoustic diagnostic. The crater diameter was of $30 \div 100$ μm; RMS of the amplitude signal was higher by 12.5% and more intense by 20% for 12Kh18N10T (AISI 321) steel than for D16 (AA 2024) alloy.

### 5.3. Further Procpects amd Paractical Significance of the Work

As was shown, the amplitude is up to 55% higher for steel and up to 25% higher for duralumin at convenient machining than 5 s before the end of processing that always stays critical for precision cutting, especially in the conditions of tool production—profiled cutters, hot channels, and injection molds. The obtained data were for the thickness of 20 mm when it stays one of the most often used thickness for EDM workpieces in tool production. The developed system proved its reliability for the samples up to 2 g when the standard sample weight for discharge gap and machining mode verifying is 15.6 g for steels and 5.4 g for aluminum for a sample of $10 \times 10$ mm in the plan with a thickness of 20 mm.

The tool wear under electrical discharge pulses has a complex character related to the thermal type of wear with a heat-affected sublayer, and the upper layer consisted of a secondary structure formed from the components of electrodes with the traces of chemical reactions at a heat of 10,000 °C. Thus, electrical discharge machining wear forms in the following stages:

- Sublimation of the electrode surfaces under discharge;
- Chemical interaction of the sublimated electrode components in the presence of high heat;
- Explosive deposition of the formed secondary structure of first and second order material;
- Re-sublimation of the secondary structure.

The explosive character of interaction between Zn and Ni should be considered while designing experiments and electrical discharge machining of chrome-nickel anti-corrosion steels. For high-precision and nano-works, machining of nickel-containing steels should be provided by a tool with no Zn in its content—copper, steel, or tungsten wire have a few disadvantages due to the softness of copper, the relatively low electrical conductivity of steels, and heat-resistance of tungsten. However, it is a promising direction for further research.

The obtained knowledge has a fundamental character and can be used as a recommendation for the industrial applications on the choice of the electrode tool material and searching the optimum EDM-factors; in this context, not only structural requirements are addressed for the working and auxiliary surfaces of the final product, but also functionality in the exploitation conditions.

**Author Contributions:** Conceptualization, S.N.G.; methodology, M.P.K.; software, K.H.; validation, P.M.P., A.N.P.; formal analysis, S.V.F.; investigation, M.P.K.; resources, P.M.P. and S.V.F.; data curation, P.A.P. and K.H.; writing—original draft preparation, A.N.P.; writing—review and editing, A.A.O.; visualization, P.A.P. and A.A.O.; supervision, M.A.V.; project administration, M.A.V.; funding acquisition, S.N.G. All authors have read and agreed to the published version of the manuscript.

**Funding:** This project has received funding from the Ministry of Education and Science of the Russian Federation within the framework of the state task for scientific research, under Grant Agreement No. 0707-2020-0025.

**Acknowledgments:** The research was done at the Department of High-Efficiency Processing Technologies of MSTU Stankin.

**Conflicts of Interest:** The authors declare no conflict of interest. The funders had no role in the design of the study; in the collection, analyses, or interpretation of data; in the writing of the manuscript, or in the decision to publish the results.

## Nomenclature

| Symbol | Description | Unit |
|---|---|---|
| $V_o$ | Operational voltage | V |
| $W_t$ | Wire tension | N |
| $I$ | Strength of the working current | A |
| $f$ | Frequency of discharge pulses | $s^{-1}$ |
| $\Delta$ | Distance between electrodes | μm |
| $\Delta_{DB}$ | Effective discharge gap | μm |
| $\Delta^*_{DB}$ | Offset of the path | μm |
| $d_w$ | Wire diameter | mm |
| $r_w$ | Wire radius | mm |
| $S_{RA}$ | Reduction area | $mm^2$ |
| $S_0$ | Original transverse area | $mm^2$ |
| $S_{min}$ | Minimal area of the final neck | $mm^2$ |
| $S_w$ | Circle segment area | $mm^2$ |
| $\alpha$ | Segment angle | degree |
| $A_n$ | Wire amplitude of $n^{th}$ vibration, n is a positive integer (1, 2, 3 ,...) | mm |
| $A_n'$ | Registered signal amplitude | mV |
| RMS | Root-mean-square mean of signal amplitude | $mV^2$ |
| $\Sigma F_{imp}$ | Summarized force of working impulses | N |
| $\Sigma E_{imp}$ | Summarized energy of working impulses | J |
| $k_n$ | Stiffness (coefficient of elasticity) | $N \cdot mm^{-1}$ |
| $m_n$ | Mass of system | g |
| $l_n$ | Wire length | mm |
| $\Delta l$ | Change in the wire length | mm |
| $T$ | Period of self-oscillations | s |
| $F_e$ | Restoring force (opposite and equal to $W_t$) | N |
| $E$ | Young's modulus | Pa |
| $R_v$ | Volumetric wear rates | $mm^3 \cdot s^{-1}$ |
| $R_m$ | Mass wear rates | $g \cdot s^{-1}$ |
| $\Delta V$ | Volumetric wear | $mm^3$ |
| $\Delta m$ | Worn mass | g |
| $t$ | Wire length wear time | s |
| $l_s$ | Slot width | mm |

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
