# Peer review of "Wire Tool Electrode Behavior and Wear under Discharge Pulses"

_technologies, doi:10.3390/technologies8030049_

Round 1
Reviewer 1 Report
- The abstract reads confusing. Please try to make the sentence structure easier in order to improve the abstract's readability.
- I would restrict the number of keywords to 5.
- Please try to avoid statements like "As it is known".
- Please try to avoid the following citation style [14-23] since the individual contributions get completely lost.
- The last 3 sections of the introduction should be shortened.
- The following sentence and statement remains rather unclear:The EDM- factors were chosen following recommendations [23,25,27,34-36]
-
The chemical compositions are given in at.-% or wt.-%?
The quality of Figure 1 is rather poor and not acceptable for a scientific article.
The same holds true for Figure 2. In addition, the data presented in Figure 2 are well known and do not need to be presented in a scientific article, since their novelty is rather limited.
The quality of Figure 3 is very poor. The same holds true for Figure 4, 5 and 6.
There is no statistical analysis. No error bars/standard deviations have been presented.
A lot of minor inconsistencies can be found throughout the entire manuscript, which greatly distract the attention of the interested reader.
The microscopic figures are also of poor quality. For SEM micrographs, please remove all instrument dependent information and just present the scale bars.
Conclusions section is too long.
Author Response
Response to Reviewer 1 Comments
Dear reviewer,
Thank you very much for your kind evaluation of our work. We do agree with all your proposals and comments and have modified the manuscript according to them.
Introduced changes were marked by green in the text of the manuscript.
Kind regards,
Authors.
Point 1: The abstract reads confusing. Please try to make the sentence structure easier in order to improve the abstract's readability.
Response 1: Thank you for your kind suggestion, the abstract is revised.
Point 2: I would restrict the number of keywords to 5.
Response 2: The keywords are revised.
Point 3: Please try to avoid statements like "As it is known".
Response 3: The text is revised.
Point 4: Please try to avoid the following citation style [14-23] since the individual contributions get completely lost.
Response 4: The citations are revised.
Point 5: The last 3 sections of the introduction should be shortened.
Response 5: The introduction is revised. These sections were a little bit shortened and simplified. Further shortening can hamper the understanding of the research filed, aims, and tasks of the study. We hope that the reviewer will find it acceptable because there is a lot of summarized data on the research that can help understand the study's key points.
Point 6: The following sentence and statement remains rather unclear:The EDM- factors were chosen following recommendations [23,25,27,34-36].
Response 6: The sentence was revised – “The EDM-factors were chosen using recommendations mentioned in previously conducted works and developed by other scientific groups [25,34-36] (Table 2).”
Point 7: The chemical compositions are given in at.-% or wt.-%?
Response 7: Thank you for this valuable comment. The percentage is given by mass fraction, i.e., the percentage by weight, in wt%.
Point 8: The quality of Figure 1 is rather poor and not acceptable for a scientific article.
Response 8: The quality of the figure was improved up to 300 dpi, the actual size is 15×11 cm for each subfigure.
Point 9: The same holds true for Figure 2. In addition, the data presented in Figure 2 are well known and do not need to be presented in a scientific article, since their novelty is rather limited.
Response 9: The figure is removed; the related fragment is revised.
Point 10: The quality of Figure 3 is very poor. The same holds true for Figure 4, 5 and 6.
Response 10: The figures were revised. The current resolution is 300 dpi for histograms and raster plots, obtained by a screenshot from the monitor of used devices, and vector plots of the obtained experimental data.
Point 11: There is no statistical analysis. No error bars/standard deviations have been presented.
Response 11: The error bars were added in Figures 5 and 6. If the reviewer reads the section of methods in detail, he or she will probably find a few passages pointing quite a large experimental pool of data.
Point 12: A lot of minor inconsistencies can be found throughout the entire manuscript, which greatly distract the attention of the interested reader.
Response 12: Of course, English is not our native language, but we expect that our English's professional level allows us to write manuscripts for publications in international scientific journals. It can be proved by the quantity and quality of our publications that our research group has published for more than 20 years. If the reviewer points particular places to improve the monotonous passages of the text and make them more harmonic and adequate from the literature point of view, we will revise these places with our best attention to the details.
However, we went through the manuscript once again and improved those passages that seem controversial to us to improve the article's readability.
Point 13: The microscopic figures are also of poor quality. For SEM micrographs, please remove all instrument dependent information and just present the scale bars.
Response 13: Our SEM images were produced with the maximum resolution possible on our SEM of the last generation. However, we have enlarged the images and showed other scale bars.
Point 14: Conclusions section is too long.
Response 14: We were astonished at this remark, as another reviewer recommended expanding the conclusion. Thus, they restructured it and tried to simplify it.

Reviewer 2 Report
The paper presents a method to understand tool electrode behavior and erosion wear in EDM process of alloys with different mechanical properties. The approach is scientifically sound and the results are well presented. It is recommended that the paper should be accepted with minor revision in terms of wording., e.g. "specially actual (P1)",...
Author Response
Thank you very much for your kind evaluation of our work. We tried to follow your recommendation and checked the manuscript once again. The mentioned collocation is revised.
Reviewer 3 Report
The paper reports about “Tool Electrode Behavior and Wear under Discharge Pulses”. The topic of the paper has a relevant interest for technology and application. However, the manuscript requires a revision prior to publication.
The following Suggestions and Comments have to be addressed before publication of the paper:
1.Please add the test stand image with descriptions.
2.In Figure 1, please describe the y axis.
3.Please add the information concerning a kind of wire (material, dimension of diameter) and a polarization of the electrodes. Also, please add some information concerning the choosing of the electrode type.
4.Please define and determine in wider range the conclusions obtained by the results analysis. Please indicate the possible causes.
5.In Figure 7 the descriptions are unreadable, please correct this.
6.In Line 253-254, “The minimal value is associated with the stable machining process and corresponds to the lowest RMS values of the signal’s amplitude”, please indicate an appropriate “the lowest RMS values of the signal’s amplitude”.
7.The images presented in Figure 11 are very valuable, please describe in wider range what has been observed and has added the results analysis.
8.In Conclusions, please add the direction of further research.
Author Response
Response to Reviewer 3 Comments
Dear reviewer,
Thank you very much for your kind evaluation of our work. We do agree with all your proposals and comments and have modified the manuscript according to them.
Introduced changes were marked by yellow in the text of the manuscript.
Kind regards,
Authors.
Point 1: Please add the test stand image with descriptions.
Response 1: The test stand image is added (Figure 2).
Point 2: In Figure 1, please describe the y axis.
Response 2: All the axes are named.
Point 3: Please add the information concerning a kind of wire (material, dimension of diameter) and a polarization of the electrodes. Also, please add some information concerning the choosing of the electrode type.
Response 3: The data on material and diameter of electrode were already presented in subsection 2.1:
The tool electrode is a brass wire with a diameter dw of 0.25 mm made of CuZn35 (Cu - 65%; Zn - 35%) with a processing temperature of 260 °C and annealing temperature of 425-750 °C.
The next was added to complete the description of the electrode and to ground the choice of material:
The choice of the electrode type was made since brass tool of 0.25 mm in diameter is the most wide-spread for wire electrical discharge machining and suitable for the broad field of applications when the forced choice of any other electrode is due to a severe technological need and is associated with the need to purchase and reinstall expensive nozzles.
It should be noted that the positive polarization of the workpiece and negative polarization of the tool electrode is traditional for the electrical discharge machining. However, modern machine tools can switch the electrodes' orientation for some particular modes or even during machining uneven and hard-to-machine materials in automatic mode.
Point 4: Please define and determine in wider range the conclusions obtained by the results analysis. Please indicate the possible causes.
Response 4: We have been confused with this recommendation since another reviewer had recommended shortening the conclusions. We have restructured them and hope that it will find the positive feedback of two reviewers. The following passages were added to the conclusions:
The tool wear under electrical discharge pulses has a complex character related to the thermal type of wear with a heat-affected sublayer, and the upper layer consisted of a secondary structure formed from the components of electrodes with the traces of chemical reactions at a heat of 10 000 °C.
The explosive character of interaction between Zn and Ni should be considered while designing experiments and electrical discharge machining of chrome-nickel anti-corrosion steels. For high-precision and nano-works, machining of nickel-containing steels should be provided by a tool with no Zn in its content – copper, steel, or tungsten wire have a few disadvantages due to the softness of copper, the relatively low electrical conductivity of steels, and heat-resistance of tungsten. However, it is a promising direction for further research.
Point 5: In Figure 7 the descriptions are unreadable, please correct this.
Response 5: Figure 7 is revised.
Point 6: In Line 253-254, “The minimal value is associated with the stable machining process and corresponds to the lowest RMS values of the signal’s amplitude”, please indicate an appropriate “the lowest RMS values of the signal’s amplitude”.
Response 6: The sentence is revised.
Point 7: The images presented in Figure 11 are very valuable, please describe in wider range what has been observed and has added the results analysis.
Response 7: Thank you for your kind suggestion. The text is revised:
Figure 11a shows the conjugation of two interdependent surfaces with two types of wear – lateral at left and front at right. The wear has a different character. The front wear surface has the appearance of the typical eroded surface – sublimated and heat-affected material coated by the secondary structure of electrode components with pores and cracks. Besides, the surface is covered by craters of solidified secondary material – usually consisted of the metallic material of the first order, solid solutions, and complex compounds of the second order (mostly oxides in the case of machining in deionized water). The craters have an explosive character that is not observed at lateral surfaces. The line of two wear surfaces conjugation is pronounced.
Figure 11b shows lateral wear at finishing. The formed surface has visible edges; the conjoined surface's left side has no presence of wear when the right side is also blank but with clear traces of secondary structure explosive deposition at the blank surface. The lateral wear surface showed two types of material destruction – the classical eroded surface of material sublimation with secondary structure deposition and mechanical wear traces.
Figure 11c shows the conjugation of two surfaces – of lateral wear and blank surface at roughing. The left side of the image – blank surface - has pronounced traces –drops, copious splashes - of explosive character of interaction occurred in the discharge gap at lateral wear. A significant volume of uneven sublimated material coated by the secondary structure with pores and cracks presents the surface with lateral wear at the right side of the image.
The front wear surface at roughing (Figure 11 d) has secondary structure pellet formation that coat the sublimated surface. The secondary sublimated surface shows typical nanoframe structure formation – more easy-to-melt material components sublimate from the secondary structure's coating(pellets) and adsorbed by the refractory matrix.
Figure 11e shows the conjugation of two surfaces – front and lateral wear at roughing. The left side of the image – front wear surface - has a coating of secondary structure.
Figure 11f shows lateral wear's surface at finishing when the obtained surface has traces of two types wear – thermal and mechanical that can be easily identified.
The cross-sections of the electrodes at roughing and finishing are presented in Figures 11g,h. Both of cross-sections showed quite intensive wear with affluent loose of the electrode material.
Point 8: In Conclusions, please add the direction of further research.
Response 8: The following passage is added to the conclusions:
The explosive character of interaction between Zn and Ni should be considered while designing experiments and electrical discharge machining of chrome-nickel anti-corrosion steels. For high-precision and nano-works, machining of nickel-containing steels should be provided by a tool with no Zn in its content – copper, steel, or tungsten wire have a few disadvantages due to the softness of copper, the relatively low electrical conductivity of steels, and heat-resistance of tungsten. However, it is a promising direction for further research.

Round 2
Reviewer 1 Report
Thanks for the revised version. The manuscript has been improved.
Specific comments:
Figure 3: Please remove the black background if possible.
Figure 5 and 6: Please remove the trends line.
Author Response
Response to Reviewer 1 Secondary Comments
Dear reviewer,
Thank you once again for your kind evaluation of our work. We do agree with your proposals and comments and have tried to modify the manuscript according to them.
Introduced changes were marked by green in the text of the manuscript.
Kind regards,
Authors.
Point 1: Figure 3: Please remove the black background if possible.
Response 1: Thank you for your kind suggestion; the figure is revised. We hope that it looks more informative and accurate now.
Point 2: Figure 5 and 6: Please remove the trends line. The second point was to remove the trend lines. In the figures, the authors present 3 data points, which is fine, but if there is no physical connection between the parameters plotted, the use of trend-lines is not appropriate.
Response 2: We tried to use this advice, but it is not entirely correct regarding graphs for EDM parameters. Typically, electrical discharge machining has a very narrow range of working factors for machining every material type. Nevertheless, sometimes more than 16 factors can be varied during machining. The papers related to EDM research concentrate on some of them as we have done regarding the research subject. For us, the importance was in the research of the wire tool's behavior under impulse, so we have decided to vary one of the most important parameters – operational voltage that influences the density of discharge pulses distribution, and wire tension that influences also wire oscillation amplitude. The detailed force diagram is presented in [1]. The main factors that influence electrical discharge machining are provided in [2].
Please, watch these examples:
- Figure 4 in [3] - work was done in the university that is in top-100 best world universities by one of the most famous research group in one of the most prospect journal; we would like to note that the provided graphs quite typical for EDM papers;
- Figure 8 in [4] – we do not know how to comment it; we used ANOVA in our research work and even published it, but when we have checked the data and compared with the experimental data we have already had, all ANOVA results were false;
- [5], [6], [7] (Figure 2 is false in wire path), [8], [11] – no even one graph on parameters; meanwhile, Metals is in Q1, Q3.
- [9] – graphs on pulse time on;
- Figure 17, [10] – it is quite difficult to comment on the provided graphs that have no axis, the authors are from the UK;
- Figure 3 [12] – histograms of roughness parameter depending on the material, no graphs of parameters;
- Figure 4,5 [13] – histograms of wear lost on the used type of machining, no graphs of parameters.
About dependences on provided factors and RMS parameters. The weight has an influence on the RMS value of signal amplitude. The amplitude of the wire can be presented by summarized force of working impulses in the system's action ΣFimp and stiffness kn that are in a relationship with mass, wire tension, and operational voltage (provided in the file).
As we have noticed, the amplitude is up to 55% higher for steel and up to 25% higher for duralumin at convenient machining than 5 s before the end of processing that always stays critical for precision cutting, especially in the conditions of tool production – profiled cutters, hot channels, and injection molds. The obtained data were for the thickness of 20 mm when it stays one of the most often used thickness for EDM workpieces in tool production. The developed system proved its reliability for the samples up to 2 g when the standard sample weight for discharge gap and machining mode verifying is 15.6 g for steels and 5.4 g for aluminum for a sample of 10×10 mm in the plan with a thickness of 20 mm. However, we have changed the graph to histograms.
The relevant passages are added to the text.
List of references
- Melnik, Y.A.; Kozochkin, M.P.; Porvatov, A.N.; Okunkova, A.A. On adaptive control for electrical discharge machining using vibroacoustic emission. Technologies 2018, 6, 96.
- Porvatov, A.N.; Kozochkin, M.P.; Fedorov, S.V.; Okunkova, A.A. About possibility of vibroacoustic diagnostics of electrical discharge machining and characterization of defects. Mech Ind. 2015, 16, 707.
- Bellotti, M.; Wu, M.; Qian, J.; Reynaerts, D. Tool Wear and Material Removal Predictions in Micro-EDM Drilling: Advantages of Data-Driven Approaches. Sci.2020, 10, 6357.
- Dzionk, S.; SiemiÄ…tkowski, M.S. Studying the Effect of Working Conditions on WEDM Machining Performance of Super Alloy Inconel 617. Machines2020, 8, 54.
- P., G.M.; G., S.; C., C.S. Experimental Investigation of Wire-EDM Machining of Low Conductive Al-SiC-TiC Metal Matrix Composite. Metals2020, 10, 1188.
- Buj-Corral, I.; Zayas-Figueras, E.; Montaña-Faiget, À. Comparative Study of Flank Cams Manufactured by WEDM and Milling Processes. Metals2020, 10, 1159.
- Abdudeen, A.; Abu Qudeiri, J.E.; Kareem, A.; Ahammed, T.; Ziout, A. Recent Advances and Perceptive Insights into Powder-Mixed Dielectric Fluid of EDM. Micromachines2020, 11, 754.
- Dong, S.; Wang, Z.; An, L.; Li, Y.; Wang, B.; Ji, H.; Wang, H. Facile Fabrication of a Superhydrophobic Surface with Robust Micro-/Nanoscale Hierarchical Structures on Titanium Substrate. Nanomaterials2020, 10, 1509.
- Machno, M. Investigation of the Machinability of the Inconel 718 Superalloy during the Electrical Discharge Drilling Process. Materials2020, 13, 3392.
- Micallef, C.; Zhuk, Y.; Aria, A.I. Recent Progress in Precision Machining and Surface Finishing of Tungsten Carbide Hard Composite Coatings. Coatings2020, 10, 731.
- Liang, X.; Liu, Y.; Ma, J.; Gong, F.; Lou, Y.; Fu, L.; Xu, B. Fabrication of Micro Ultrasonic Powder Molding Polypropylene Part with Hydrophobic Patterned Surface. Materials2020, 13, 3247.
- Mouralova, K.; Zahradnicek, R.; Benes, L.; Prokes, T.; Hrdy, R.; Fries, J. Study of Micro Structural Material Changes after WEDM Based on TEM Lamella Analysis. Metals2020, 10, 949.
- Martynenko, V.; Martínez Krahmer, D.; Nápoles Alberro, A.; Cabo, A.; Pérez, D.; Zayas Figueras, E.E.; Gonzalez Rojas, H.A.; Sánchez Egea, A.J. Surface Damaging of Brass and Steel Pins when Sliding over Nitrided Samples Cut by Finishing and Roughing EDM Conditions. Materials2020, 13, 3199.
